# Persistent epigenetic memory impedes rescue of the telomeric phenotype in human ICF iPSCs following DNMT3B correction

Shir Toubiana[1,2†], Miriam Gagliardi[3†‡], Mariarosaria Papa[3], Roberta Manco[3], Maty Tzukerman[1,2], Maria R Matarazzo[3*], Sara Selig[1,2*]

[1]Molecular Medicine Laboratory, Rappaport Faculty of Medicine, Technion, Haifa, Israel; [2]Rambam Health Care Campus, Haifa, Israel; [3]Institute of Genetics and Biophysics, ABT CNR, Naples, Italy

**Abstract** DNA methyltransferase 3B (DNMT3B) is the major DNMT that methylates mammalian genomes during early development. Mutations in human *DNMT3B* disrupt genome-wide DNA methylation patterns and result in ICF syndrome type 1 (ICF1). To study whether normal DNA methylation patterns may be restored in ICF1 cells, we corrected *DNMT3B* mutations in induced pluripotent stem cells from ICF1 patients. Focusing on repetitive regions, we show that in contrast to pericentromeric repeats, which reacquire normal methylation, the majority of subtelomeres acquire only partial DNA methylation and, accordingly, the ICF1 telomeric phenotype persists. Subtelomeres resistant to de novo methylation were characterized by abnormally high H3K4 trimethylation (H3K4me3), and short-term reduction of H3K4me3 by pharmacological intervention partially restored subtelomeric DNA methylation. These findings demonstrate that the abnormal epigenetic landscape established in ICF1 cells restricts the recruitment of DNMT3B, and suggest that rescue of epigenetic diseases with genome-wide disruptions will demand further manipulation beyond mutation correction.

**\*For correspondence:**
maria.matarazzo@igb.cnr.it (MRM);
seligs@technion.ac.il (SS)

[†]These authors contributed equally to this work

**Present address:** [‡]Max Planck Institute of Psychiatry, Translational Psychiatry, Munich, Germany

**Competing interests:** The authors declare that no competing interests exist.

## Introduction

CpG methylation is an essential chemical modification involved in the regulation of DNA accessibility to the transcriptional machinery (*Luo et al., 2018*). Changes in the DNA methylation landscape may drastically alter gene expression and downstream phenotypes (*Dor and Cedar, 2018*; *Luo et al., 2018*). Thus, establishment of DNA methylation patterns during development is strictly regulated. De novo DNA methylation is catalyzed by DNA methyltransferases 3A and 3B (DNMT3A and DNMT3B), which methylate many genomic sites redundantly (*D'Antonio et al., 2012*). However, DNMT3B is the chief DNMT in charge of de novo methylation in embryos (*Huntriss et al., 2004*), and the majority of repetitive regions are exclusively methylated by DNMT3B (*Kato et al., 2007*; *Okano et al., 1999*). While there are indications that de novo methylation is primarily regulated at the level of DNMT3B recruitment (*Baubec et al., 2015*), the exact rules that govern the establishment of methylation patterns are yet unclear. Several of the factors suggested to influence DNMT3B recruitment are chromatin modifications, non-coding RNAs, and binding of RNA polymerase II and specific transcription factors (*Baubec et al., 2015*; *Gatto et al., 2017*; *Greenfield et al., 2018*; *Hackett and Surani, 2013*; *Smith and Meissner, 2013*).

Knockout of *Dnmt3b* in mice leads to embryonic lethality (*Okano et al., 1999*; *Ueda et al., 2006*). In humans, *DNMT3B* is one of four genes that lead to the rare autosomal recessive ICF (Immunodeficiency, Centromeric instability and Facial anomalies) syndrome when mutated

(*von Bernuth et al., 2014*; *de Greef et al., 2011*; *Thijssen et al., 2015*). All ICF subtypes exhibit DNA hypomethylation at various genomic regions, including pericentromeric satellite 2 and 3 repeats, that consequently fail to condense properly during mitosis (*Xu et al., 1999*). However, ICF1, caused by bi-allelic loss of function mutations in *DNMT3B* (always with some residual activity), differs from the remaining ICF subtypes by its relatively unperturbed centromeric methylation and striking hypomethylation at subtelomeric regions (*Toubiana et al., 2018*). The mechanistic basis for this phenotypic distinction between ICF syndrome subtypes is yet undeciphered.

Subtelomeres, the regions immediately adjacent to telomeres, consist of CpG-rich repetitive DNA (*Cross et al., 1990*; *de Lange et al., 1990*; *Stong et al., 2014*) that undergoes DNA methylation during early development (*de Lange et al., 1990*). While in several organisms the telomeric and subtelomeric chromatin share similar heterochromatic marks with centromeric and pericentromeric regions, the chromatin status at human telomeres and subtelomeres is less clear (*Galati et al., 2013*). Approximately two thirds of human subtelomeres include at close proximity to the TTAGGG-telomeric repeats CpG-rich promoters that transcribe the long non-coding RNA TERRA (*Diman and Decottignies, 2018*). In ICF1 syndrome, severe subtelomeric hypomethylation is accompanied by elevated levels of TERRA transcripts and accelerated telomere shortening that leads to premature senescence in ICF1 fibroblasts (*Yehezkel et al., 2008*; *Yehezkel et al., 2013*). Certain telomeres are more prone to shorten in ICF1 cells (*Sagie et al., 2017a*), suggesting that when DNMT3B is defective, subtelomeric-specific characteristics contribute in cis to the degree of telomere loss.

The molecular abnormalities in ICF1 embryos are anticipated to initiate during the first days of development, when DNMT3B, the major DNMT present and active at that stage (*Borgel et al., 2010*; *Hackett and Surani, 2013*; *Uysal et al., 2015*), fails to properly methylate its targets. Human induced pluripotent stem cells (iPSCs) mimic this stage in development (*Davidson et al., 2015*) and therefore are a potent tool to study methylation abnormalities occurring in developing ICF embryos. Indeed, fibroblast-like cells (FLs) derived from ICF1 iPSCs, recapitulated the abnormal telomeric phenotype and entered premature senescence due to accelerated telomere shortening (*Sagie et al., 2014*).

Ectopic expression of WT DNMT3B in ICF1 fibroblasts fails to rescue hypomethylation at either pericentromeric repeats or at subtelomeres (*Yehezkel et al., 2013*). One explanation for this scenario is that conditions permitting de novo methylation of repetitive sequences are restricted to the implantation stage. Here, we utilize ICF1 iPSCs to analyze the factors that regulate subtelomeric methylation during early development. To this end, we generated isogenic ICF1 iPSCs with and without *DNMT3B* mutations by genome editing. We show that, despite the restored methylation capacity of DNMT3B, the majority of subtelomeric regions did not reacquire normal DNA methylation due to restricted binding of the corrected DNMT3B. Our data demonstrate that several factors influence the capacity to recruit DNMT3B to subtelomeres, including abnormal TERRA transcription and trimethylation of H3K4 (H4K4me3). Accordingly, pharmacological reduction of H3K4me3 levels allowed partial restoration of subtelomeric DNA methylation. Thus, rescuing the abnormal DNA methylation patterns in ICF1 syndrome requires additional manipulations beyond mutation correction.

## Results

### CRISPR/Cas9 correction of *DNMT3B* mutations in ICF iPSCs restores the catalytic activity of DNMT3B

Homology-directed repair (HDR) by CRISPR/Cas9 editing was implemented to rescue the catalytic activity of DNMT3B in two iPSCs derived from ICF1 patients pR and pG (*Sagie et al., 2014*) (all further reference to ICF patients or cells relates to ICF1 only) (*Figure 1—figure supplement 1*). We aimed to correct at least one allele in each ICF iPSC, as carriers for ICF syndrome have a normal phenotype (*Sagie et al., 2017a*; *Sagie et al., 2017b*; *Yehezkel et al., 2008*). Following this procedure, we identified two homozygous corrected clones for pR iPSCs, cR7 and cR35, obtained at passages 24 and 27, respectively. For pG iPSCs we identified two corrected clones, cG13 and cG50, at passages 15 and 17, respectively. These two clones had corrected the mutation leading to a premature stop codon in *DNMT3B* and maintained a missense mutation in the catalytic domain of the protein (*Figure 1—figure supplement 1*). Western analysis indicated that DNMT3B protein levels both prior

and post correction in ICF iPSCs do not differ substantially from those of WT iPSCs (*Figure 1—figure supplement 2*).

To determine whether correction of *DNMT3B* restored the catalytic capacity of DNMT3B, we studied the methylation status of pericentromeric satellite 2 repeats by Southern analysis of samples digested with a methylation sensitive restriction enzyme. Hypomethylation is evident in the ICF

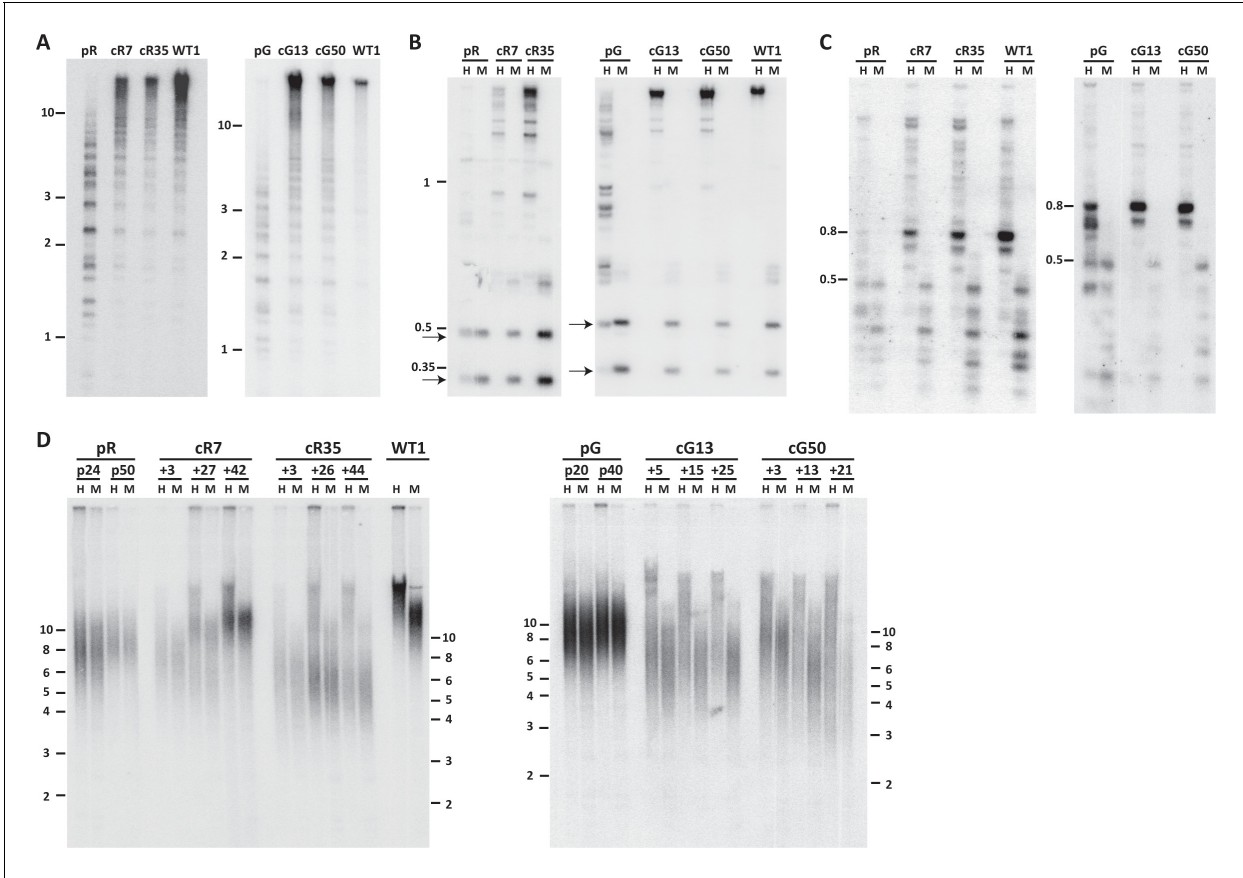

**Figure 1.** Pericentromeric regions are methylated de novo following *DNMT3B* correction, while subtelomeres are only partially methylated in corrected-ICF iPSCs. Following correction of at least one allele of *DNMT3B* in ICF iPSCs pR and pG (see *Figure 1—figure supplement 1*), the activity of the restored DNMT3B was determined, as shown in (**A–D**). (**A**) DNA methylation of pericentromeric satellite 2 repeats determined by Southern analysis. DNA from ICF iPSCs (pR and pG), their corrected clones (cR7, cR35 and cG13, cG50, respectively) and WT iPSCs UN1-22 (WT1) was digested with the methylation-sensitive restriction enzyme *BstBI* and hybridized to a satellite 2 probe. Size markers in kilobases (kb) appear on the left. Additional analysis of satellite 2 repeats was done by targeted bisulfite analysis (*Figure 1—figure supplement 3B*). (**B**) Methylation analysis of pericentromeric NBL-1 repeat. DNA extracted at similar passages as in (**A**) was digested with either *MspI* (M) or *HpaII* (H). The presence of the two lower molecular bands in the *HpaII*-digested DNA (arrows) indicates hypomethylation of this repeat. Additional analysis of NBL1 repeats, as well as other pericentromeric repeats was carried out by WGBS analysis (*Figure 1—figure supplement 3A*, *Figure 1—figure supplement 4*). (**C**) Methylation analysis of pericentromeric p1A12 repeat. DNA extracted at similar passages as in (**A**) was digested with either *MspI* (M) or *HpaII* (H). Hybridization bands below 0.5 kb in the *HpaII*-digested DNA indicate that this repeat is hypomethylated. Additional analysis of p1A12 repeats, as well as other pericentromeric repeats was carried out by WGBS analysis (*Figure 1—figure supplement 3A*, *Figure 1—figure supplement 4*). (**D**) Methylation analysis of subtelomeric regions. DNA of ICF iPSCs pR and pG, their corrected clones cR7, cR35 and cG13, cG50, respectively, and WT iPSCs UN1-22 (WT1) was digested with the isoschizomeric *MspI* (M) and *HpaII* (H) enzymes at three time points following corrected colony isolation. The passage at which DNA was extracted for analysis appears above the lanes. The plus (+) indicates the number of passages since the corrected clones were isolated. Southern analysis was performed using a C-rich telomeric probe, as described (*Yehezkel et al., 2008*). Size markers in kb appear both on the left and on the right to depict the degree of separation at each side.

The online version of this article includes the following figure supplement(s) for figure 1:

**Figure supplement 1.** CRISPR/Cas9–mediated *DNMT3B* correction in ICF iPS cells.

**Figure supplement 2.** Expression of catalytically active isoforms DNMT3B1 and DNMT3B2 in WT and isogenic ICF iPSCs.

**Figure supplement 3.** Restored DNA methylation at pericentromeric repeats.

**Figure supplement 4.** Effect size values measured for pericentromeric repeats.

iPSCs by the presence of low molecular weight bands, as seen for pR and pG (*Figure 1A*). All four corrected clones, cR7, cR35, cG13 and cG50 (three to four passages after clone isolation), showed a significant shift of hybridization to higher molecular weight bands, indicating that these regions had undergone significant de novo methylation early after correction (*Figure 1A*). Rapid de novo methylation following *DNMT3B* correction was apparent in two additional repeat families, NBL-1 and p1A12 which are located at proximity to centromeric regions of acrocentric chromosomes (*Brock et al., 1999*; *Thoraval et al., 1996*) (*Figure 1B and C*).

We additionally utilized high throughput sequencing-based approaches to measure single CpG-methylation levels at the various pericentromeric repeats. In agreement with the Southern analysis, whole genome bisulfite sequencing (WGBS) indicated that pericentromeric repeats were remethylated after *DNMT3B* correction (*Figure 1—figure supplement 3A*). Effect size comparison analysis of DNA methylation levels at the various repeats demonstrated that the corrected ICF iPSCs were highly similar to the WT iPSCs (*Figure 1—figure supplement 4*). Furthermore, to determine the de novo methylation dynamics over time, we performed targeted sequencing of a chromosome 1-specific satellite 2 amplicon generated from bisulfite treated DNA, at 3 and 23 passages following correction (*Figure 1—figure supplement 3B*). This analysis indicated that the majority of methylation was acquired rapidly after editing, however following further passaging, these repeats acquired some additional methylation. Based on the regain of methylation at pericentromeric repeats we could conclude that the methyltransferase activity of DNMT3B was rescued following HDR correction.

## Restored DNMT3B partially remethylates subtelomeres

We next examined whether methylation was restored also at subtelomeric regions. We first analyzed the methylation for all subtelomeres collectively by Southern analyses. Samples from three time points following correction were compared to the original ICF iPSCs at two passages, as well as to control iPSCs (*Figure 1D*). DNA was digested with the isoschizomeric methylation-sensitive and non-sensitive restriction enzymes (*HpaII* and *MspI*, respectively) and hybridized with a telomeric probe (*Yehezkel et al., 2008*). In the two original ICF iPSCs the *MspI* and *HpaII* hybridization patterns were almost identical, indicating severe subtelomeric hypomethylation. In contrast, the *HpaII* hybridization smear in the corrected clones was elevated to significantly higher molecular weights in comparison to the *MspI* smear. This revealed that a substantial fraction of subtelomeric regions regained methylation that inhibited *HpaII* digestion and generated longer restriction fragments. However, while the WT iPSC *HpaII* digested sample showed a compacted smear with no low molecular weight hybridization signals, in all corrected clones the *HpaII* hybridization smear spread down to low molecular weights, indicating that not all subtelomeric regions regained methylation following correction. To note, the various corrected clones showed variable telomere length dynamics over extended culturing periods with no clear relationship to the degree of subtelomeric remethylation they underwent. However, iPS clonal variability with respect to telomere length has been described previously (*Sagie et al., 2014*; *Suhr et al., 2009*; *Vaziri et al., 2010*; *Yehezkel et al., 2011*).

## De novo DNA methylation at subtelomeres is tightly regulated

To evaluate the DNA methylation status at individual subtelomeres following *DNMT3B* correction, we used WGBS data to determine the mean DNA methylation of all CpG sites along the most distal 2 kb of 24 individual subtelomeres (*Figure 2A*). A marked difference in subtelomeric methylation levels was apparent when comparing WT iPSC to either of the ICF iPSCs (pR and pG). However, subtelomeres clearly varied with respect to their degree of hypomethylation in ICF iPSCs prior to correction. Strikingly, subtelomere clustering, based on the degree of methylation regained following *DNMT3B* correction, clearly indicated that methylation was also differentially restored to each subtelomere. By ranking the subtelomeres according to the degree of methylation acquired following correction (*Figure 2B*), a positive correlation was evident between the original degree of methylation in the ICF iPSCs, and the capacity to acquire methylation following correction. The pattern of remethylation was distinctly shared among the four corrected clones, suggesting that subtelomeric de novo methylation is a tightly regulated process.

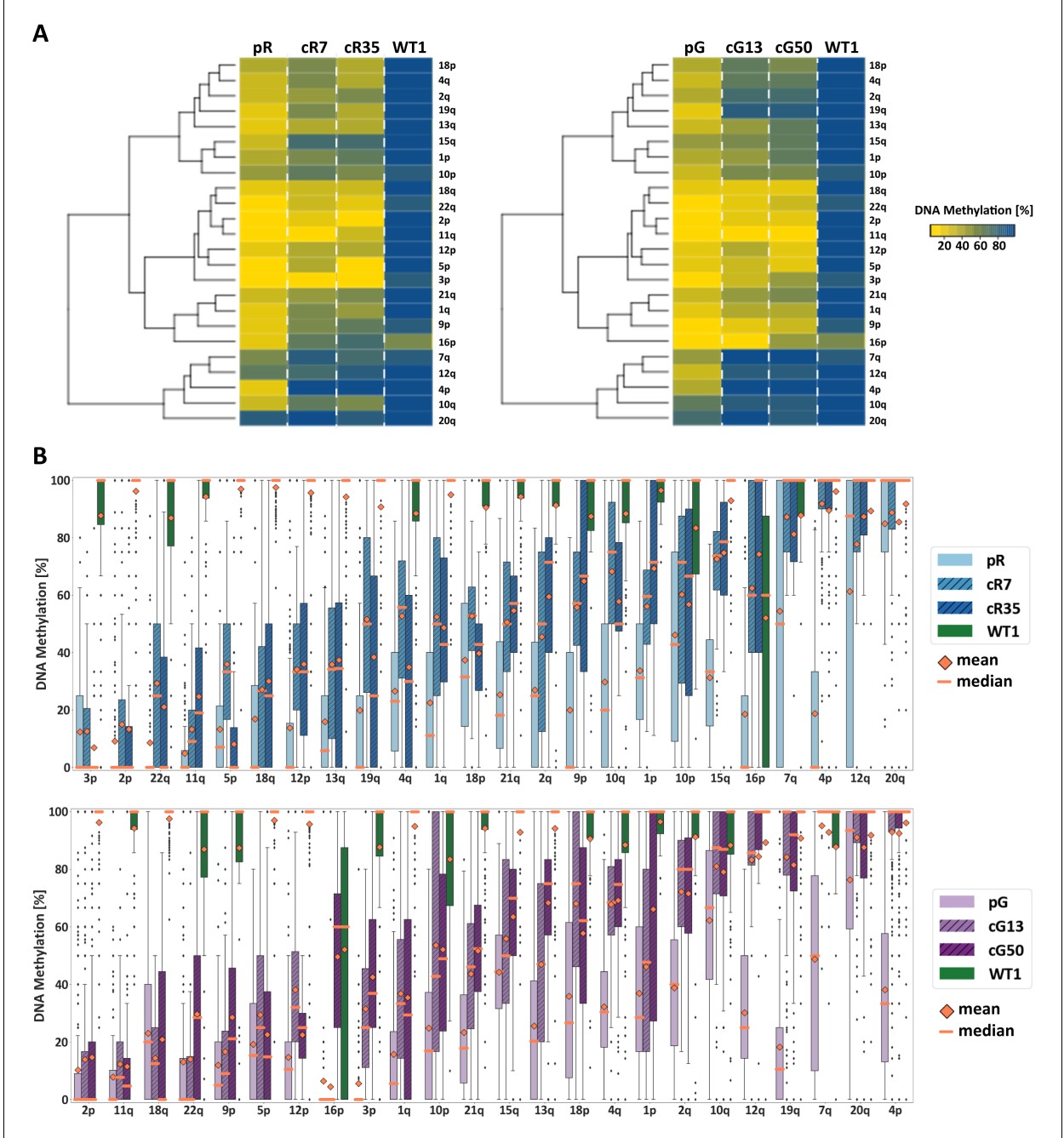

**Figure 2.** Subtelomeric methylation is differentially restored following *DNMT3B* correction. (A) Subtelomeric methylation analysis by whole genome bisulfite sequencing (WGBS). DNA of ICF iPSCs pR and pG, their corrected clones cR7, cR35 and cG13, cG50 respectively, and WT iPSCs UN1-22 (WT1) was subjected to WGBS sequencing. DNA from corrected clones was analyzed 35 and 12 passages following correction of pR and pG iPSCs, respectively. The heatmaps display the mean methylation of all CpGs within the most distal 2 kb prior to the telomere tract of 24 subtelomeres. Hierarchical clustering of samples appears to the left of the heatmaps. (B) DNA methylation level distribution of the analyzed subtelomeres based on the degree of restored methylation following correction. Methylation level distribution is presented by boxplots for each of the 24 analyzed subtelomeres, with mean and median values indicated as orange diamonds and bars, respectively. Upper plot - pR and corrected clones, lower plot - pG and corrected clones. Subtelomeres are ranked from left to right based on averaging of the methylation level medians of both corrected clones.

## Remethylation at TERRA promoters is correlated with *DNMT3B* binding capacity

We next evaluated whether subtelomeric remethylation is influenced by context-dependent sequences in the most distal 2 kb of human subtelomeres, including TERRA promoter elements. The canonical TERRA promoter is composed of three family repeats of 61-, 29- and 37 bps (*Brown et al., 1990*; *Nergadze et al., 2009*) which are fully or partially shared by the various TERRA promoters (*Figure 3A* and *Figure 3—figure supplement 1*).

We first utilized WGBS data to scrutinize the methylation status along the distal 2 kb of various subtelomeres in relation to the CpG density and the position of the TERRA promoter repeats (*Figure 3—figure supplement 1*). When focusing on regions with high CpG density, it was evident that subtelomeres varied in the degree of methylation acquired following correction and there was no clear association between CpG density and the capacity to regain DNA methylation at subtelomeres. Some subtelomeres remained notably hypomethylated, such as 1q, 2p, 5p and 11q. Many subtelomeres regained partial methylation, such as 2q, 10q, 13q and 21q. Only subtelomere 4p restored DNA methylation to normal levels in all corrected clones. When we examined in detail the methylation status of the TERRA promoter repeats, we found that all three repeat families were substantially protected from de novo methylation following correction (*Figure 3B* and *Figure 3—figure supplement 2*).

To study the methylation dynamics of TERRA promoters at specific subtelomeres with high sequencing coverage and at different time points after *DNMT3B* correction, we performed targeted high throughput sequencing of subtelomere-specific amplicons generated from bisulfite converted DNA. Despite the low complexity of these regions, we succeeded in amplifying regions of TERRA promoters from 14 subtelomeres (*Figure 3A*). Methylation analysis of these regions confirmed that subtelomeric regions differed based on their capacity to regain methylation (*Figure 3C* and *Figure 3—figure supplement 3*). Several of the tested amplicons (5p, 2p, 13q, 1q/21q, 2q/4q) remained drastically hypomethylated, even after prolonged passaging (reaching up to 54 passages in the case of cR7).

Conversely, other subtelomeric regions (9p/Xq, 2q/4q/10q, and 10q) regained intermediate to high levels of methylation. Notably, regions in 4p, 10p/18p and 7q promoters, gradually acquired normal levels of methylation in all corrected clones. Similar to the analysis of the entire 2 kb subtelomeric regions (*Figure 2*), specific subtelomeric regions that were initially more methylated in ICF iPSCs regained methylation more readily in comparison to severely hypomethylated regions. When comparing all corrected clones, we could observe that cG13 and cG50 regained methylation at a faster rate than cR7 and cR35, even though pR iPSCs were corrected at both mutated alleles in comparison to only one allele in pG iPSCs. However, in general, the four corrected clones behaved very similarly, reinforcing the former conclusion that methylation at subtelomeres is conducted in a highly regulated manner.

Previous studies demonstrated that the capacity of a specific genomic region to undergo de novo methylation was dependent on DNMT3B recruitment (*Baubec et al., 2015*). To test whether the degree of restored methylation was correlated with DNMT3B binding, we determined the enrichment of DNMT3B at individual subtelomeres by chromatin immunoprecipitation (ChIP) (*Figure 3D*). Indeed, ChIP revealed that TERRA promoter regions with persisting hypomethylation in ICF iPSC, such as 2p, 5p and 11q, were poorly enriched for DNMT3B binding in comparison to WT iPSCs, even following correction. Conversely, the binding of the corrected DNMT3B at promoter regions that partially or fully regained DNA methylation (10q/13q/19q and 7q, respectively) (*Figure 3C* and *Figure 3—figure supplement 1*) was comparable to the binding of the WT DNMT3B in the control iPSC. As expected, the fully remethylated region of satellite 2 in the corrected clones was highly enriched with DNMT3B. Based on the combined data describing DNA methylation and DNMT3B binding at TERRA promoters, we could conclude that the capacity to undergo de novo subtelomeric methylation by the corrected DNMT3B was clearly correlated with DNMT3B recruitment.

## High TERRA levels persist following *DNMT3B* correction

We previously demonstrated that TERRA levels are significantly elevated in ICF iPSCs (*Sagie et al., 2014*). The persisting hypomethylation of at least several TERRA promoters in the corrected ICF

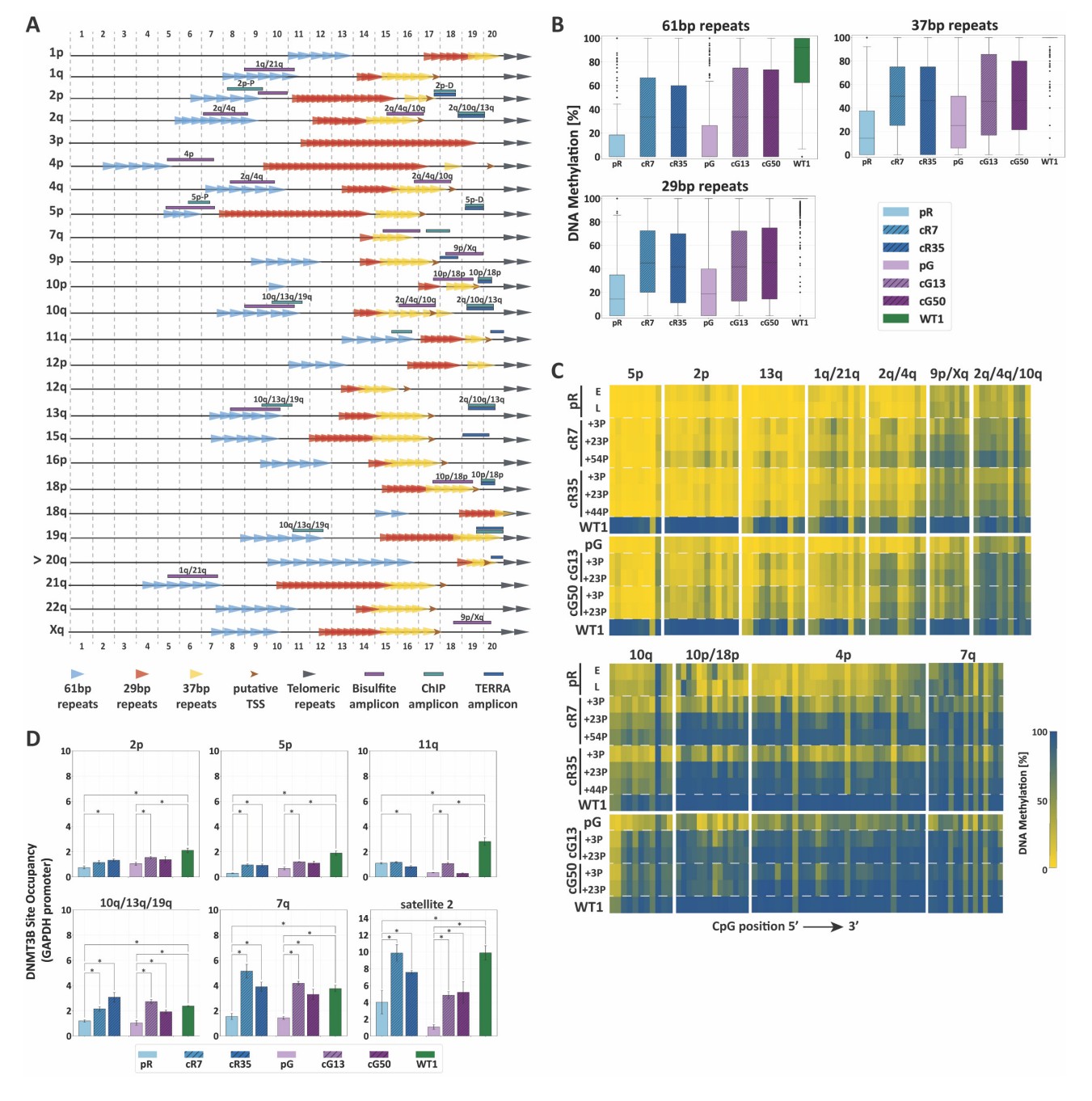

**Figure 3.** TERRA promoter repeats are variably resistant to de novo methylation following *DNMT3B* correction in correlation with DNMT3B enrichment. (**A**) TERRA promoter components along human subtelomeres. The positions of the 61-, 29- and 37 bp repeats comprising TERRA promoters are depicted along the distal 2 kb of a subset of human telomeres. Putative transcription start sites (TSS) are based on *Diman et al. (2016)*. The distal ends of the subtelomeres are depicted by the gray arrows on the right. The arrow to the left of telomere 20q indicates that this telomere lacks a clear telomeric tract at its distal end. The regions amplified in the various analyses, are indicated on the maps. The 2 kb regions are divided into 100 bps bins, marked at both top and bottom of the maps. TERRA promoter elements on additional subtelomeres appear in *Figure 3—figure supplement 1*. (**B**) Distribution of DNA methylation levels at the TERRA promoter repeats. The boxplots demonstrate the DNA methylation levels as determined by WGBS of the 61-, 29- and 37 bp repeats from many subtelomeres collectively, in ICF iPSCs pR and pG, their corrected clones cR7, cR35 and cG13, cG50 respectively, and WT iPSCs (WT1) (See also *Figure 3—figure supplements 1* and *2* and *Supplementary file 2*. (**C**) DNA methylation at specific TERRA promoter regions following *DNMT3B* correction. Targeted bisulfite analysis of specific TERRA promoter regions from 14 subtelomeres in ICF iPSC, their corrected clones at two or three time points following isolation, and WT UN1-22 iPSCs (WT1). The passage (P) at which DNA was extracted for analysis appears to the left of the heatmaps. The plus (+) indicates the number of passages since the corrected clones were isolated. The heatmaps display the

*Figure 3 continued on next page*

*Figure 3 continued*

methylation percentage across the various amplicons. Each subtelomeric region consists of several columns, each representing a specific CpG site, in the 5′ to 3′ direction of the sequence (left to right). (See also *Figure 3—figure supplement 3* and *Supplementary file 1*. (D) DNMT3B binding at various subtelomeres prior and post correction. Corrected clones were analyzed at passages 35–45 following isolation. Subtelomeres 2p and 5p were analyzed with the upstream 2p-P and 5p-P primer sets (*Figure 3A*, and *Supplementary file 3*. Two-tailed Mann-Whitney U-tests were performed to determine statistical differences between WT and ICF samples and between the original ICF iPSCs and their corrected clones (*=p value<0.05). Bars and error bars represent means and SEM of at least three experimental repeats.

The online version of this article includes the following figure supplement(s) for figure 3:

**Figure supplement 1.** DNA methylation profile of the 2 kb distal regions of several subtelomeres.
**Figure supplement 2.** Effect size values measured for subtelomeric TERRA promoter repeats.
**Figure supplement 3.** Methylation percentage distribution of all CpG sites per amplicon for ICF, corrected-ICF and WT iPSCs.

IPSCs, suggested that TERRA would continue to be upregulated from a subgroup of subtelomeres. To confirm this assumption, we determined TERRA levels at individual telomere ends in ICF iPSCs at early and late time points following *DNMT3B* correction by subtelomere-specific quantitative RT-PCR (RT-qPCR) (*Figure 4*). We examined 12 subtelomeres with clear TERRA promoters as well as subtelomere 7q that has only a few promoter elements (*Diman et al., 2016*; *Sagie et al., 2017b*)

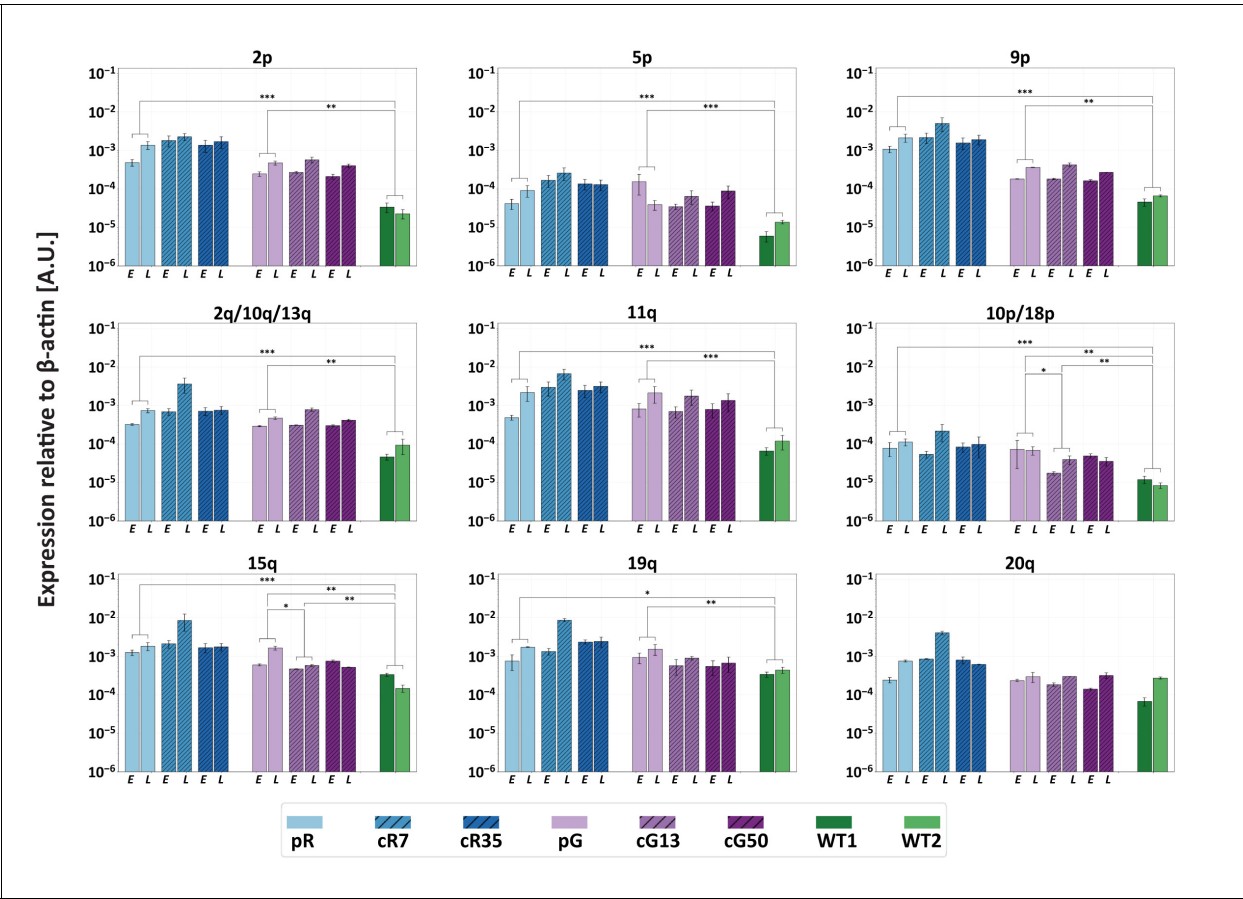

**Figure 4.** Restoration of DNMT3B activity is insufficient to repress TERRA expression. TERRA expression from various telomeres depicted above the graphs, was determined for uncorrected ICF iPSCs, corrected clones and control WT iPSCs (UN1-22 and FSE-5m; WT1 and WT2, respectively), using primers described in *Supplementary file 3*. TERRA was determined for the ICF iPSCs at an early (E) and late (L) passage. The time points at which TERRA levels were determined: pR- passages 30 and 66, cR7 – 14 and 54 passages following isolation, cR35 –14 and 47 passages following isolation, pG- passages 27 and 44, cG13 – 10 and 30 passages following isolation, cG50 – 7 and 28 passages following isolation, WT1 – passage 48, WT2 – passage 64. Each bar represents the mean of the relative TERRA expression compared to expression of β-actin in the same sample. Error bars represent SEM of at least three experimental repeats. A one-tailed Mann-Whitney U-test was performed to determine statistical differences between WT and ICF samples and between the original ICF and its corrected clones (*=p value<0.05, **=p value<0.01, ***=p value<0.001).

(*Figure 3A*). In ICF iPSCs prior to correction, no 7q-TERRA transcripts were detected, while the remaining telomeres transcribed TERRA at various levels, with pR iPSCs expressing in general higher TERRA levels in comparison to pG iPSCs. In all cases, except telomere 20q, significant differences in TERRA levels were apparent between the WT and original ICF iPSCs. Following *DNMT3B* correction, we found that, with exception of 15q and 10p/18p in cG13, TERRA expression was not altered, even

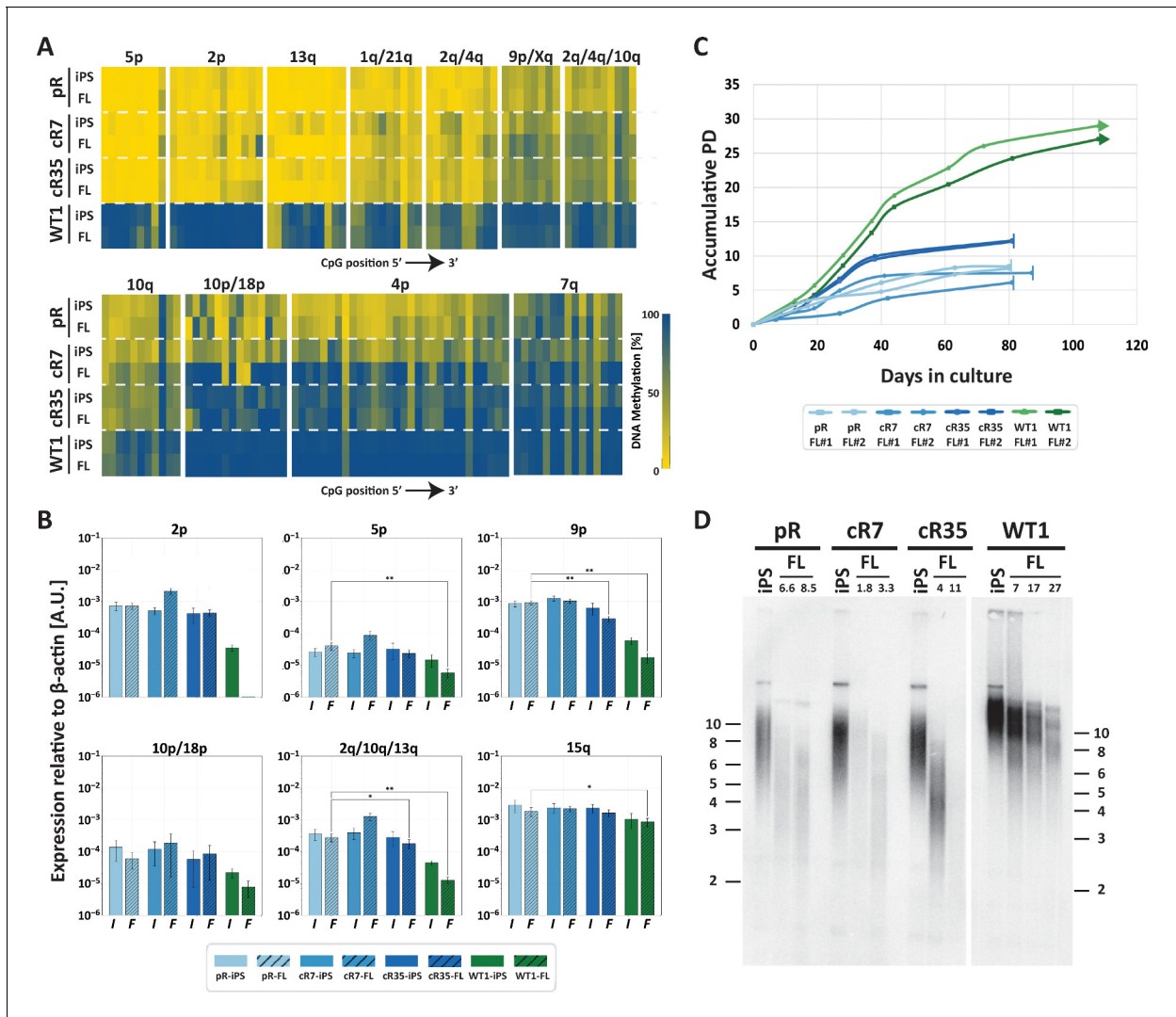

**Figure 5.** Fibroblast-like cells derived from *DNMT3B*-corrected ICF iPSCs enter premature senescence. (A) Subtelomeric DNA methylation in fibroblast-like cells (FLs) derived from *DNMT3B*-corrected ICF iPSCs. DNA methylation was determined as described in *Figure 3C*, for iPSCs and FLs of pR, cR7, cR35 and WT1, using primers reported in *Supplementary file 1*. The analyzed amplicons for targeted bisulfite sequencing are depicted in the map in *Figure 3A*. (B) TERRA expression levels at nine subtelomeres were determined by RT-qPCR of iPSCs and their derived clones, using primers reported in *Supplementary file 3*. Samples were compared by a one-tailed Mann-Whitney U-test (*=p value<0.05, **=p value<0.01, ***=p value<0.001). Bars and error bars represent means and SEM of at least three experimental repeats. No transcription was detected for subtelomere 2p in WT FLs, therefore no values are displayed for this sample. (C) Growth curves of pR, cR7, cR35 and WT1-derived FLs. The FLs of pR, cR7 and cR35 were passaged twice until senescence (depicted by a vertical line at the end of the growth curve), and the WT1 FLs were passaged until approximately PD 30 and were still actively dividing at that point (depicted by an arrow at the end of the growth curve). Senescence was also determined by positive SA-β-GAL staining (*Figure 5—figure supplement 1*). (D) Telomere lengths of the original iPSCs and their derived FLs were determined by Terminal Restriction Fragment (TRF) analysis. PDs at which samples were analyzed appear above the lanes of FLs. Size markers in kb appear both on the left and on the right to depict the degree of separation at each side.

The online version of this article includes the following figure supplement(s) for figure 5:

**Figure supplement 1.** FLs derived from *DNMT3B*-corrected ICF iPSCs display the SA-β-Gal senescence marker at an abnormally low PD.

**Figure supplement 2.** Ectopic expression of hTERT rescues premature senescence of corrected ICF FLs.

at late passages. Notably, despite the significant reduction in TERRA levels emanating from 15q and 10p/18p in cG13, they remained significantly higher in comparison to the WT iPSCs. Thus, restoring the catalytic activity of DNMT3B did not reduce the abnormally high TERRA expression, at least from the tested subset of telomeres.

## Fibroblast-like cells derived from the corrected ICF iPSCs enter premature senescence

Based on the persisting TERRA expression in the corrected ICF iPSCs, we speculated that telomerase-negative differentiated derivatives of these iPSCs would experience accelerated telomere shortening, similar to the original ICF fibroblasts. To this end, we differentiated pR iPSCs, its corrected cR7 and cR35 clones at 3 and 23 passages following correction, respectively, as well as WT iPSCs. We then isolated fibroblast-like (FLs) cells and passaged them continuously while documenting their population doublings (PDs). The passages at which we differentiated cR7 and cR35 were chosen based on the similar telomere lengths of both clones at those time points. We first examined the subtelomeric methylation status in the iPSCs compared to the derived FLs (*Figure 5A*) and found that cR35 shows no significant changes in methylation levels at most subtelomeres. The methylation changes between cR7 iPSCs and FLs were more noticeable compared to cR35, probably because this clone was differentiated at a very early time point following correction, when most subtelomeres had scarcely regained methylation. The persisting presence of DNMT3B1, the catalytically active isoform of DNMT3B, during the first stages of embryoid body (EBs) formation, could lead to additional subtelomeric methylation before this active isoform was silenced in the terminally differentiated FLs (*Sagie et al., 2014*; *Yehezkel et al., 2011*). Nevertheless, either significant (5p, 2p, 13q, 1q/21q, 2q/4q) or intermediate (9p/Xq, 2q/4q/10q and 10q) subtelomeric hypomethylation persisted in the FLs of both corrected clones (*Figure 5A*). Accordingly, even though TERRA levels at several subtelomeres were partially diminished in ICF FLs compared to the undifferentiated ICF iPSCs, the subtelomeres that were highly expressed at the undifferentiated state (2p, 5p, 9p, 2q/10q/13q and 15q), remained highly expressed in the ICF FLs in comparison to the WT FLs (*Figure 5B*).

Two separate FLs clones derived from each of the corrected pR iPSCs, recapitulated the abnormal premature senescence documented in the original ICF fibroblasts (*Yehezkel et al., 2013*). These FLs arrested their growth (*Figure 5C*) and were positive for SA-β-Gal staining at a very low PD of between 6–12 PDs (*Figure 5—figure supplement 1*). In comparison, control WT were still actively dividing at PD 30. Analyzing telomere lengths in the various FLs indicated that premature senescence occurred in ICF FLs due to highly accelerated telomere shortening, even in those derived from the corrected ICF iPSCs (*Figure 5D*). Ectopic expression of telomerase in FLs derived from corrected iPSC clones, significantly extended the proliferative capacity of these cells, supporting accelerated telomere shortening as the instigator of premature senescence (*Figure 5—figure supplement 2*). Based on the telomeric phenotype of the corrected FLs, we conclude that partially restoring subtelomeric methylation is insufficient for rescuing the premature senescence phenotype in ICF FLs.

## Subtelomeric de novo DNA methylation is inhibited by H3K4me3 enrichment

We next explored whether chromatin modifications may hinder recruitment of DNMT3B to subtelomeric regions in the corrected ICF iPSCs. We observed that already during the process of reprogramming ICF fibroblasts into iPSCs, subtelomeres differed in their capacity to undergo partial de novo methylation (*Figure 6A*). We also found that abnormally high TERRA expression in ICF fibroblasts (*Figure 6B*) was associated with H3K4me3 enrichment at subtelomeres (*Figure 6C*). The H3K4me3 modification is strongly associated with open transcribed chromatin, inhibits DNMT3B binding (*Baubec et al., 2015*; *Greenfield et al., 2018*; *Morselli et al., 2015*; *Rose and Klose, 2014*; *Weber et al., 2007*), and is enriched at subtelomeres in ICF LCLs (*Deng et al., 2010*). This suggested that H3K4me3 patterns set at subtelomeres in ICF fibroblasts prevailed throughout the process of reprogramming into iPSC and influenced the capability of the mutated DNMT3B (and perhaps of the WT DNMT3A) to methylate subtelomeric regions.

Interestingly, the subtelomeres that were partially de novo methylated during reprogramming to iPSCs corresponded to those which regained methylation following editing of *DNMT3B* (*Figure 3C*).

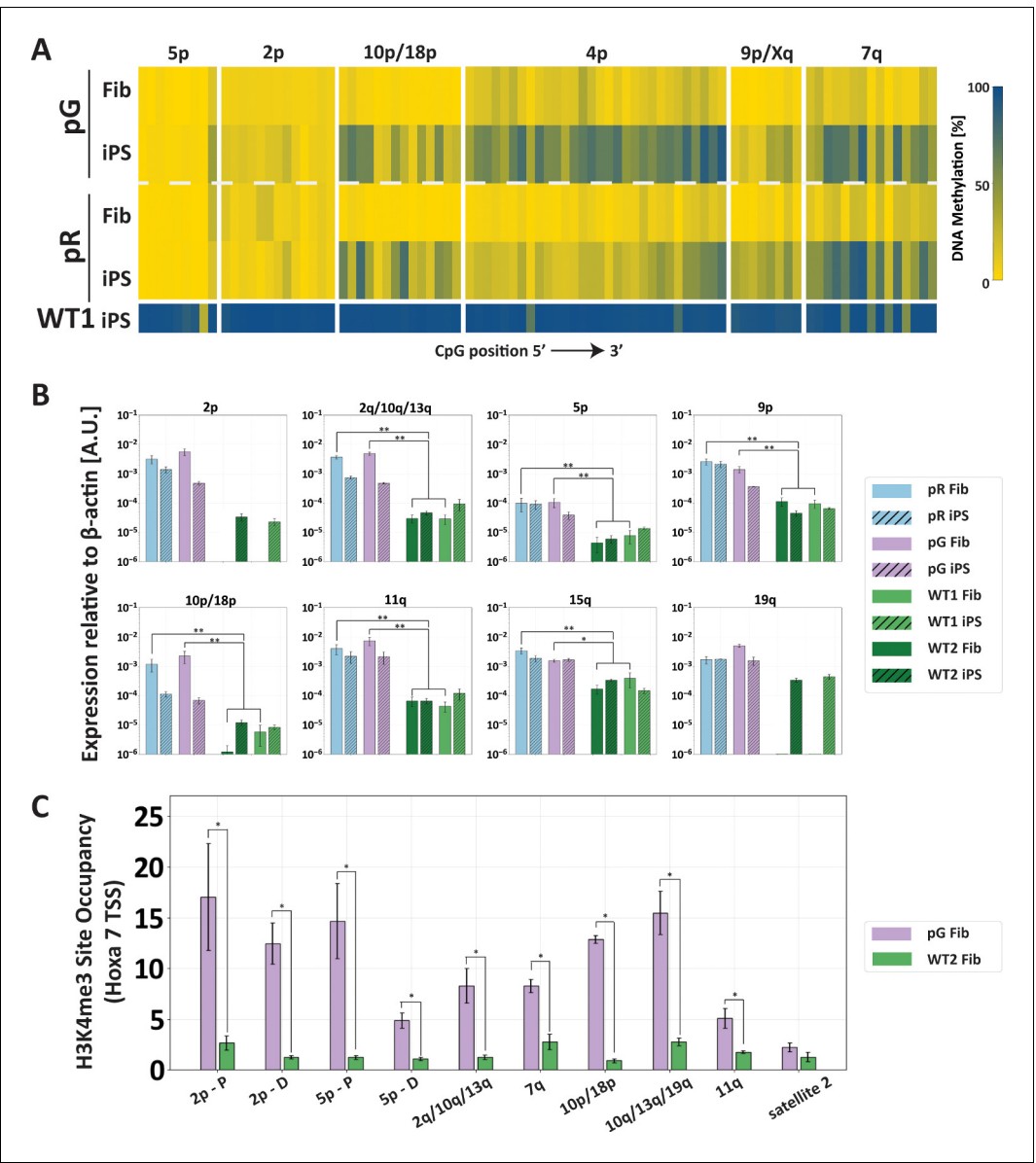

**Figure 6.** The abnormal epigenetic landscape at subtelomeres is set in the original ICF fibroblasts. (A) Subtelomeres are de novo methylated to various degrees during reprogramming of ICF fibroblasts into iPSCs. Methylation analysis of various subtelomeres indicates that during reprogramming subtelomeres were partially de novo methylated. However, subtelomeres in ICF iPSCs remained markedly hypomethylated in comparison to WT iPSCs. DNA methylation was determined as described in *Figure 3C*, using primers reported in *Supplementary file 1*. The analyzed amplicons for targeted bisulfite sequencing are depicted in the map in *Figure 3A*. (B) TERRA is highly expressed in ICF fibroblasts. TERRA expression levels were compared between fibroblasts and the generated iPSCs in pR, pG and two normal WT controls, UN1-22 (WT1) and FSE-5m (WT2). TERRA levels for each subtelomere are expressed relative to β-actin expression in the sample. No transcription was detected for subtelomeres 2p and 19q in WT fibroblasts, therefore no values are displayed for these samples. TERRA expression levels were compared between samples by a two-tailed Mann–Whitney U-test (*=p-value<0.05, **=p-value<0.01). Bars and error bars represent means and SEM of at least three experimental repeats. (C) H3K4me3 enrichment at selected subtelomeres in ICF (pG) and WT (WT2) fibroblasts. ChIP amplicon regions are reported in the map in *Figure 3A*. Fold change in site occupancy at each specific region was determined in relation to the negative control region Hoxa 7 TSS amplified in the same sample (*=p-value<0.05, two-tailed Mann–Whitney U-test). Bars and error bars represent means and SEM of at least three experimental repeats. Primer sets used for ChIP analysis are reported in *Supplementary file 3*.

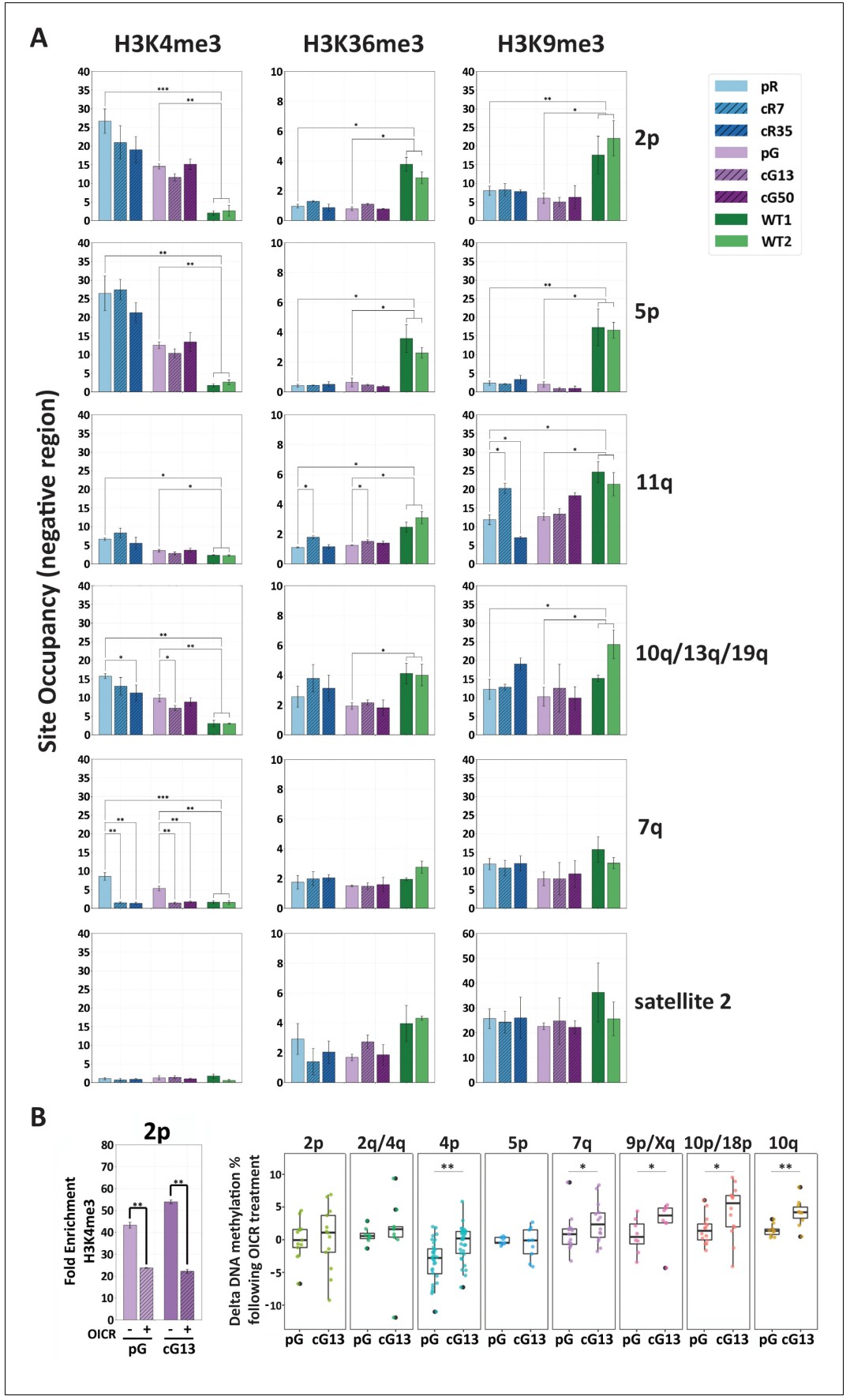

**Figure 7.** H3K4me3 enrichment influences de novo methylation at subtelomeric regions. (**A**) Enrichment for H3K4me3, H3K36me3 and H3K9me3 at subtelomeres was determined by ChIP assay for ICF original and corrected iPSCs and WT iPSCs. Enrichment of these histone marks was assayed at TERRA promoters depicted on the right, as well as at satellite 2 repeats. Subtelomeres 2p and 5p were analyzed with the upstream 2p-P and 5p-P primer sets (*Figure 3A* and *Supplementary file 3*. Fold change in site occupancy was determined in relation to the following control regions amplified in the same samples: Hoxa7 TSS, Myoglobin exon 2 and GAPDH promoter for H3K4me3, H3K36me3 and H3K9me3 marks, respectively (See also *Figure 7—figure supplement 1*). Bars and error bars represent means and SEM of at least three experimental repeats. (**B**) Pharmacological inhibition of H3K4 MLL methyltransferase with OICR-9429 treatment. Left panel: H3K4me3 enrichment at subtelomere 2p (analysis of region 2p-P) determined by ChIP in uncorrected (pG) and corrected (cG13) iPSCs, either with (+) or without (-) OICR-9429 treatment. Fold enrichment was calculated by normalizing the percentage of input to the IgG negative control. Right panel: Boxplots showing the distribution of delta values calculated by comparing the CpG methylation levels obtained by targeted bisulfite sequencing in treated *vs.* untreated samples for subtelomeres 2p, 2q/4q, 4p, 5p, 7q, 9p/Xq, 10p/18 p and 10q. Note that a unique primer set, designated by [d] in *Supplementary file 1*, was used in this experiment to amplify subtelomeres 10q, 2q and 4q. Statistical analysis for ChIP was performed using a two-tailed Student's t-test. Statistical analysis of methylation data was performed by two-tailed Mann-Whitney U-tests (*=p value<0.05, **=p value<0.01).

The online version of this article includes the following figure supplement(s) for figure 7:

**Figure supplement 1.** Chromatin marks at proximal and distal regions of specific subtelomeres.

We therefore investigated the effect of several chromatin modifications on de novo DNA methylation in the corrected ICF iPSCs by ChIP (*Figure 7A*). We first determined whether H3K4me3 was enriched in corrected ICF iPSCs and found that this modification persisted following *DNMT3B* correction in the majority of the analyzed subtelomeres. This was most evident at the proximal regions of subtelomeres 2p, 5p as well as 11q, although at this latter subtelomere the H3K4me3 enrichment appeared generally lower. Other subtelomeres, such as 10q/13q/19q, showed low but significant reduction in this histone mark in two corrected clones, however, the level of this modification still remained higher in comparison to WT iPSCs. The reduction in H3K4me3 binding in corrected clones was more evident at regions in close proximity to the telomere tract (*Figure 7—figure supplement 1*). The non-transcribing subtelomere 7q lost all H3K4me3 enrichment following correction, in correlation with its remethylation to WT levels. Satellite 2 repeat was not enriched for this modification in either of the iPSCs. This correlated with lack of transcriptional activity of satellite 2 in all samples (results not shown) and reacquisition of DNA methylation to normal levels following correction.

H3K36me3 is an additional histone mark shown to recruit DNMT3B1 to active gene bodies in mouse stem cells (*Baubec et al., 2015*; *Neri et al., 2017*). H3K36me3 and H3K4me3 are mutually exclusive (*Morselli et al., 2015*; *Zhang et al., 2015*) and accordingly, TERRA promoters in subtelomeres 2p, 5p and 11q showed very low enrichment of H3K36me3 in all ICF iPSCs, whether corrected or not (*Figure 7A* and *Figure 7—figure supplement 1*). In other assayed regions, no significant differences were found in H3K36me3 between the isogenic ICF iPSCs (*Figure 7A* and *Figure 7—figure supplement 1*). The subtelomeric regions downstream to the TERRA transcription start sites (TSS) were not enriched with H3K36me3 (*Figure 7—figure supplement 1*), indicating that in contrast to gene bodies of most active genes, even when TERRA is highly transcribed (such as in the case of 2p and 2q/10q/13q) the TERRA template regions are not marked with this modification.

H3K9me3 is a histone mark highly associated with heterochromatin (*Machida et al., 2018*). Conflicting data exists regarding the role of this mark in directing Dnmt3b to major satellite repeats in mouse embryonic stem (ES) cells (*Déjardin, 2015*; *Lehnertz et al., 2003*). We assayed H3K9 trimethylation to determine whether it plays a role in recruiting DNMT3B to human subtelomeres. Apart from subtelomere 11q in one corrected clone, our ChIP analysis found no significant differences in enrichment of H3K9me3 in ICF iPSCs prior and post *DNMT3B* correction (*Figure 7A* and *Figure 7—figure supplement 1*). Subtelomeres 2p and 5p, which were markedly resistant to de novo methylation, were significantly less enriched for H3K9me3 in ICF iPSCs relative to WT iPSCs. TERRA-less subtelomere 7q, which regained normal methylation following correction, was enriched for H3K9me3 in all iPSC types.

Collectively, our ChIP data suggested that H3K4me3 enrichment in ICF iPSCs prior to *DNMT3B* correction, plays a major role in determining the capacity of a subtelomere to recruit DNMT3B. In line with this view, we attempted to pharmacologically reduce H3K4me3 enrichment in ICF iPSCs by

utilizing the small molecule OICR-9429 that inhibits the MLL methyltransferase/WDR5 interaction (*Grebien et al., 2015*). As prolonged treatment with OICR-9429 leads to considerable cell death, we could subject the cells to this small molecule for 72 hr, after which we confirmed that this short treatment reduced H3K4me3 enrichment at subtelomeres, as demonstrated for subtelomere 2p (*Figure 7B*, left panel). We then compared the percentage of DNA methylation values obtained by targeted bisulfite sequencing of several subtelomeres in OICR treated *vs.* untreated samples of both corrected (cG13) and uncorrected (pG) ICF iPSCs and found higher positive delta values in the majority of subtelomeres in the corrected iPSCs, where DNMT3B activity was restored. Subtelomeres highly enriched with H3K4me3, such as 2p and 5p, showed a less marked increase in DNA methylation compared to the other subtelomeres, probably due to the short exposure to the drug (*Figure 7B*, right panel). Altogether, these data strongly support that high enrichment of the H3K4me3 mark at subtelomeres in ICF iPSCs prevents the recruitment of the corrected DNMT3B.

## Discussion

The major DNMT that de novo methylates DNA during implantation is DNMT3B (*Huntriss et al., 2004*). DNA methylation plays crucial roles during development, as reflected by the lethality of mouse embryos lacking this enzyme (*Okano et al., 1999*; *Ueda et al., 2006*). In humans, biallelic loss of function mutations in *DNMT3B* lead to ICF syndrome which displays a wide range of phenotypes arising as a consequence of genome-wide methylation perturbations. The abnormal methylation patterns in this syndrome have been described so far in a limited number of cell types (*Gatto et al., 2017*; *Heyn et al., 2012*; *Huang et al., 2014*; *Simo-Riudalbas et al., 2015*; *Velasco et al., 2018*), and the molecular chain of events that links the abnormal DNA methylation with the majority of the phenotypes in this disease is still unclear.

De novo methylation is predominantly regulated by recruitment of DNMT3B to its target sites (*Baubec et al., 2015*). Several factors have been implicated in regulation of DNMT3B recruitment (*Baubec et al., 2015*; *Gatto et al., 2017*; *Greenfield et al., 2018*; *Hackett and Surani, 2013*; *Smith and Meissner, 2013*), however, the exact rules that direct, or conversely, inhibit DNMT3B binding, are still not completely resolved. In this study we took advantage of iPSCs derived from ICF fibroblasts to advance our understanding of these rules. ICF iPSCs provide a cell system that on the one hand mimics the embryonic stage at which de novo methylation occurs, including the expression of the full length DNMT3B1 isoform, and on the other hand has an inherently hypomethylated genome. Correction of the mutated *DNMT3B* gene in these cells by genome editing, allowed us to trace events of remethylation at various genomic regions. Comparing the hypomethylated starting point with the methylation status following *DNMT3B* correction is advantageous over the opposite approach where DNMT3B is deleted in pluripotent stem cells (*Horii et al., 2013*; *Liao et al., 2015*), as the removal of DNMT3B from normally methylated human embryonic stem cells does not substantially alter genome-wide DNA methylation (*Liao et al., 2015*). This may be attributed to a limited role of DNMT3B in methylation maintenance in iPSCs compared to de novo methylation. The comparison of isogenic ICF iPSCs prior and post *DNMT3B* correction, enabled the precise identification of de novo methylation events that could be attributed solely to the recovered enzyme activity. Using such comparisons, we verified that very early following isolation of the corrected clones, satellite 2 hypomethylation, a hallmark of ICF syndrome, was reversed and these regions reached methylation levels highly comparable to those in WT iPSCs (*Figure 1A* and *Figure 1—figure supplement 3*). De novo methylation of pericentromeric repeats (*Figure 1A–C* and *Figure 1—figure supplement 3*) not only confirmed that the catalytic activity of DNMT3B was restored, but also that, in contrast to somatic cells (*Yehezkel et al., 2013*), iPSCs provide a cellular milieu that can support de novo methylation.

To ask whether additional genomic regions restored normal DNA methylation following the rescue of DNMT3B, we focused our attention on subtelomeric regions which are also drastically hypomethylated in *DNMT3B*-mutated ICF patients, and are associated with a distinct telomeric phenotype (*Deng et al., 2010*; *Yehezkel et al., 2008*; *Yehezkel et al., 2013*). However, telomeric regions differ from pericentromeric regions in their epigenetic marks (*Negishi et al., 2015*; *Rosenfeld et al., 2009*), and defects in centromeric, pericentromeric and subtelomeric regions vary among subtypes of ICF syndrome (*Toubiana et al., 2018*), pointing to disparities in DNA methylation regulation between these various classes of repetitive regions. This suggested that the dynamics

of methylation acquisition following *DNMT3B* correction may follow a different course at subtelomeres compared to pericentromeric repeats. Moreover, even among human subtelomeres, despite sharing high sequence similarity, variations exist in GC skewing, in CpG density and in presence of TERRA promoters (*Sagie et al., 2017b*). These properties were proposed to contribute to variability in subtelomeric hypomethylation, in DNA:RNA hybrid levels (*Sagie et al., 2017b*), as well as in the propensity of telomeres to shorten in ICF syndrome cells (*Sagie et al., 2017a*) and could also affect the capability to restore DNA methylation. Indeed, the current study further emphasizes the variability among human subtelomeres, as not all subtelomeres regained methylation at the same rate and to the same degree following correction of *DNMT3B*.

Surveying the subtelomeres following correction clearly illustrated the partial methylation rescue of these regions, even following numerous passages in culture. Strikingly, the four corrected clones, originating from two patient iPSCs, carrying different *DNMT3B* mutations, demonstrated consistent patterns of remethylation at the majority of subtelomeres, highlighting the tight regulation of this process. The various subtelomeres could be roughly divided into three categories: those that acquired methylation readily, those that gradually regained methylation over time, and those that persistently remained hypomethylated even after prolonged passaging in culture. Noticeably, subtelomeres from the first group were those that displayed initially higher methylation levels in ICF cells (*Figures 2* and *3C*) and also tended to undergo a certain degree of de novo methylation during the generation of iPSCs from the ICF fibroblasts (*Figure 6A*). These subtelomeres may either be characterized by chromatin that is more accessible to DNMT3B, whether mutated or not, or may be partially recognized by DNMT3A, which is also expressed in pluripotent stem cells (*Liao et al., 2015*), albeit at much lower levels, and may partially compensate for the lack of DNMT3B activity.

Previous studies had suggested that de novo methylation is regulated by DNMT3B recruitment (*Baubec et al., 2015*) and our findings confirmed the correlation between DNMT3B binding and the capacity to undergo methylation at subtelomeres (*Figure 3D*). The differential reacquisition of methylation at various subtelomeres provided us with a unique opportunity to identify factors in cis and *trans* that support or restrain DNMT3B binding at these regions. We postulated that transcriptional activity and associated chromatin modifications play a significant role in determining the accessibility of a specific subtelomeric region to DNMT3B. Analysis of TERRA expression revealed that most of the TERRA promoters were highly active both prior and post correction, and this transcriptional activity persisted also at late passages following correction (*Figure 4*). However, several subtelomeres which do express TERRA, such as 10p/18p and 2q/4q/10q, underwent full or partial DNA methylation following correction (*Figure 3C*) suggesting that TERRA expression solely could not explain the failure to recruit DNMT3B to subtelomeric regions.

Transcriptionally active promoters are usually marked by H3K4 trimethylation (*Buratowski and Kim, 2010*), an epigenetic mark shown to inhibit recruitment of DNMT3B (*Morselli et al., 2015*). Enrichment of H3K4me3 at TERRA promoters, was found to be correlated with TERRA levels in human fibroblasts and cancer cell lines (*Caslini et al., 2009*; *Negishi et al., 2015*), and this was also the case for subtelomeres in ICF fibroblasts and iPSCs (*Figures 6C* and *7A*). We postulated that this enrichment would persist at a subset of TERRA promoters even after *DNMT3B* editing and prevent the recruitment of the corrected DNMT3B. Indeed, subtelomeres that remained hypomethylated following correction were found in most cases to remain significantly enriched with H3K4me3 in comparison to WT iPSCs (*Figure 7A*). Consistently, when we partially reduced H3K4me3 levels at subtelomeres in corrected ICF iPSC clones by pharmacological interference, we found an increase in subtelomeric methylation that is less pronounced at highly H3K4me3 enriched subtelomeres such as 2p and 5p (*Figure 7B*).

Thus, we could conclude that transcriptional activity at subtelomeres and H3K4 trimethylation are factors that discourage DNMT3B binding to subtelomeres and therefore prevent the restoration of the normal DNA methylation patterns following DNMT3B correction. Nevertheless, some subtelomeres did not completely adhere to these rules. Subtelomere 11q stands out in this respect, demonstrating that despite relatively low H3K4me3 levels (*Figure 7A*), it remained severely hypomethylated following correction (*Figure 3—figure supplement 1*). Notably, TERRA expression from 11q remained significantly high compared to WT iPSCs, even in the corrected clones. Together this suggests the existence of additional unidentified factors that play roles in directing de novo methylation at subtelomeres and remain to be identified in future studies.

Telomeres provide protection to chromosome ends and their epigenetic properties are essential for their maintenance and function (*Galati et al., 2013*). Since premature senescence is induced even by a few dysfunctional telomeres (*Hemann et al., 2001*; *Ouellette et al., 2000*; *Steinert et al., 2000*; *Zou et al., 2004*), the persisting hypomethylation at a subset of subtelomeres following *DNMT3B* correction, prevented the rescue of the telomeric phenotype in the FLs generated from the corrected ICF iPSCs (*Figure 5*). The failure to reacquire normal methylation patterns following rescue of DNMT3B's catalytic capacity, and the consequential lack of phenotypic rescue, illustrate the challenges in treating genetic diseases that affect the entire epigenome. In the case of ICF syndrome, although the primary defect which initiates during implantation is manifested as abnormal DNA methylation, ICF somatic cells, such as the fibroblasts from which the iPSCs were established, display in addition to the disrupted DNA methylation patterns, abnormal histone modifications and transcriptional activity, at least as much as demonstrated at telomeric regions. These abnormal features prevail in the ICF iPSCs and consequentially, the epigenetic landscape encountered by the corrected DNMT3B is significantly altered in comparison to that present during normal embryonic development. A question arises whether the ability to overcome the persisting epigenetic memory and fully rescue the methylation defects in ICF cells may be feasible in stem cells corresponding to earlier developmental stages (*Davidson et al., 2015*; *Liu, 2017*). Reversion of primed human stem cells to a naïve state is accompanied by a significant global reduction in DNA methylation and H3K9me3 levels (*Pastor et al., 2016*; *Takashima et al., 2014*; *Wang et al., 2018*), thus rendering the chromatin to a more open conformation. Specifically at CpG rich promoters, an increase in active histone modifications (H3K4me2/3) was apparent in naïve stem cells (*Wang et al., 2018*). As we show that the high H3K4me3 levels constitute a major barrier to DNA methylation recovery at subtelomeres, we do not anticipate that a shift to a more naïve state will assist in overcoming the permissive ground state of ICF syndrome chromatin, which is impeding the binding of the corrected DNMT3B. However, we cannot exclude that H3K4me3 levels may show a distinct profile at TERRA promoters in the naïve state, and in addition, that a varying repertoire of transcription factors could alleviate the permissive epigenetic landscape, allowing the binding of DNMT3B. Exploring this possibility awaits future studies in which corrected ICF iPSCs will be reverted into a more naïve state. Nevertheless, the success in treating patients with genome-wide epigenetic perturbations, will require additional steps beyond the correction of the genetic mutation responsible for initiation of the disease. Specifically for ICF syndrome, we demonstrated the proof of principle that such a strategy, as temporary removal of H3K4me3, may partially recover DNA methylation at subtelomeres.

In this study we largely focused on the DNA methylation dynamics at subtelomeric regions. We envision that future genome-wide studies of the epigenetic characteristics in ICF iPSCs prior and post *DNMT3B* correction will further decode the rules that dictate DNMT3B recruitment to chromatin. Understanding these rules will contribute to future design of therapy for ICF syndrome patients, and will be relevant to additional pathological states, such as cancer, in which normal DNA methylation patterns are likewise perturbed.

## Materials and methods

**Key resources table**

| Reagent type (species) or resource | Designation | Source or reference | Identifiers | Additional information |
|---|---|---|---|---|
| Gene (human) | *DNMT3B1* | www.ensembl.org | ENST00000328111.6 | |
| Cell line (human; Female – pR, cR7, cR35; Male – pG, cG13, cG50, UN1-22, FSE-5m) | iPSC lines | This paper; *Shinnawi et al., 2015*; *Sagie et al., 2014*; *Yehezkel et al., 2011* | | Cell lines maintained in S. Selig and M. Matarazzo labs |
| Cell line (human; Female – pR, cR7, cR35; Male – pG, cG13, cG50, UN, FSE) | Fibroblasts and Fibroblast-like cells (FLs) | This paper; *Sagie et al., 2014* | | Cell lines maintained in S. Selig lab |

*Continued on next page*

*Continued*

| Reagent type (species) or resource | Designation | Source or reference | Identifiers | Additional information |
|---|---|---|---|---|
| Antibody | H3K4me3 (Rabbit polyclonal) | Abcam, Cambridge, UK | Cat# ab8580 | ChIP (3.5 µg per sample) |
| Antibody | H3K9me3 (Rabbit polyclonal) | Abcam, Cambridge, UK | Cat# ab8898 | ChIP (3.5 µg per sample) |
| Antibody | H3K36me3 (Rabbit polyclonal) | Abcam, Cambridge, UK | Cat# ab9050 | ChIP (5 µg per sample) |
| Antibody | anti-DNMT3B (Rabbit polyclonal) | Diagenode | Cat# C15410218 | ChIP (5 µg per sample) Western (1:3000) |
|  | anti-DNMT3B (Rabbit polyclonal) | Diagenode | Cat# C15410218 | Western (1:1000) |
| Antibody | anti-actin (Rabbit polyclonal) | Sigma | Cat# a2066 | Western (1:5000) |
| Recombinant DNA reagent | pSpCas9 (BB)—2A-Puro (PX459) V2.0 | Addgene plasmid #62988; http://www.addgene.org/62988/; RRID:Addgene_62988 | *Ran et al., 2013* | sgRNAs were designed using the Optimized CRISPR Design online tool (http://crispr.mit.edu) and cloned into this plasmid |
| Recombinant DNA reagent | lentiviral vector pLOX-TERT-iresTK | Addgene plasmid # 12245; http://www.addgene.org/12245/; RRID:Addgene_12245 | *Salmon et al., 2000* |  |
| Sequenced-based reagent | pR screening primers | This paper (IDT) | PCR primers | Forward 5'-ATGGCGAG GGCAGAAAGAGT-3', Reverse- 5'-GCACCGTG TTAGGCTGCTCC-3' |
| Sequence-based reagent | pG screening primers | This paper (IDT) | PCR primers | Forward- 5'-GCCTGTCCA CATGGAACC-3', Reverse- 5'-CTAGAGCTC TCTTCCCTCC-3' |
| Sequenced-based reagent | The template ssODNs of pR | This paper (IDT) | Oligonucleotides | pR: 5'- ACGCTCCAGG ACCTTCCCA GCAGCTTCTGGCGGGC ACCACGGCCCATGTTGG ACACGTCTGTATAGTGC ACAGGAAAGCCAAAGATCCTG-3' |
| Sequenced-based reagent | The template ssODNs of pG | This paper (IDT) | Oligonucleotides | pG: 5'-CGCTCACTATGTCAGCG CCTGACACCACCCCAGTCCCCA CTGCCCAGCCACCTCTGA TCTCGGGGGTCCGGATAG CCTCCAGGATTGGGG-3' |
| Peptide, recombinant protein | protease inhibitor cocktail | MERCK, Darmstadt, Germany | Cat. #: 11697498001 |  |
| Peptide, recombinant protein | proteinase K | MERCK, Darmstadt, Germany | Cat. #: 03115828001 |  |
| Peptide, recombinant protein | RNA purification with TURBO DNA-*free* TM Kit | Thermo Fisher Scientific | Cat. #: AM2238 |  |
| Peptide, recombinant protein | SuperScript III reverse transcriptase | ThermoFisher Scientific | 18080044 |  |
| Commercial assay or kit | DNA bisulfite-conversion | EPIGENTEK, NY, USA | Methylamp DNA Modification kit |  |

*Continued on next page*

Continued

| Reagent type (species) or resource | Designation | Source or reference | Identifiers | Additional information |
|---|---|---|---|---|
| Commercial assay or kit | DNA amplification using FastStart Taq DNA Polymerase, dNTPack | MERCK, Darmstadt, Germany | Cat. #: 04738314001 | |
| Commercial assay or kit | Magna ChIP | MERCK, Darmstadt, Germany | Cat. #: 17–610 | |
| Commercial assay or kit | Senescence beta-Galactosidase Staining Kit | Cell Signaling Technology | Cat. #: 9860S | |
| Chemical compound, drug | ROCK inhibitor | StemCell Technologies | Cat. #: Y-27632 | (5 µM) |
| Chemical compound, drug | L-755507 | StemCell Technologies | Cat. #: 73992 | *Yu et al., 2015* (5 µM) |
| Chemical compound, drug | Accutase | StemCell Technologies | Cat. #: 07920 | |
| Chemical compound, drug | recombinant Human FGF-basic | PeproTech | Cat. #: 100-18B | |
| Chemical compound, drug | Opti-MEM | ThermoFisher Scientific | Cat. #: 31985070 | |
| Chemical compound, drug | Puromycin | MERCK, Darmstadt, Germany | Cat. #: P7130 | (1 µg/ml) |
| Chemical compound, drug | OICR-9429 | MERCK Darmstadt, Germany | Cat. #: SML1209 | dissolved in DMSO (1 µM) |
| Chemical compound, drug | Fast SYBR Green Master Mix | ThermoFisher Scientific | Cat. #: 4385612 | |
| Chemical compound, drug | Phenyl methanesulfonyl fluoride | MERCK, Darmstadt, Germany | Cat. #: P7626 | (1 mM) |
| Software, algorithm | Python libraries: 'numpy', 'scipy', 'statsmodels', 'pandas', 'jupyter' and 'notebook' | Python | | Python 2.7 |
| Software, algorithm | ImageQuant software | GE Healthcare Life Science | | |
| Software, algorithm | FastQC tool | https://www.bioinformatics.babraham.ac.uk/projects/fastqc/ | | |
| Software, algorithm | Bismark | https://www.bioinformatics.babraham.ac.uk/projects/bismark/ | | |
| Software, algorithm | BLAT software | University of California, Santa Cruz [UCSC] Genome Browser, http://genome.ucsc.edu | | |
| Software, algorithm | MAFFT v7 | https://mafft.cbrc.jp/alignment/software/ | | |
| Software, algorithm | Repitools R package | http://bioconductor.org/packages/release/bioc/html/Repitools.html | | |
| Software, algorithm | heatmap2 function in R 3.03 | https://www.rdocumentation.org/packages/gplots/versions/3.0.1.1/topics/heatmap.2 | | |

*Continued on next page*

Continued

| Reagent type (species) or resource | Designation | Source or reference | Identifiers | Additional information |
|---|---|---|---|---|
| Software, algorithm | Bedtools v2.17.0 | https://github.com/arq5x/bedtools2 | | |

## Cell culture of induced pluripotent cells (iPSCs)

IPSCs (*Table 1*) were plated on tissue culture plates coated with Cultrex Reduced Growth Factor Basement Membrane Matrix (343300501, Trevigen, MD, USA), and grown in either mTeSR-1 media (85850, StemCell Technologies, Vancouver, Canada) or PeproGrow hESC Media (hESC-100, Pepro-Tech, NJ, USA). Cells were passaged every 4–6 days. Cell were dissociated from plates with 0.5 mM EDTA and following each passage, 2 µM Thiazovivin (14245, Cayman CHEMICAL, MI, USA was added to the media.

FLs derived from all types of iPSCs (see below) were grown in fibroblast media (DMEM media supplemented with 20% fetal bovine serum, 2 mM glutamine, 100 U/ml penicillin and 100 µg/ml streptomycin) at standard conditions. Every 4–8 days, cells were sub-cultured to maintain a continuous log phase growth and the PD at each passage was determined as described (*Yehezkel et al., 2013*).

The identity of iPSCs, FLs and fibroblasts was authenticated by Sanger sequencing of the patient-specific mutations and of the silent mutations incorporated during the editing process into the *DNMT3B* gene. All cells were negative in a screen for mycoplasma.

## CRISPR/Cas9 mediated HDR of ICF iPSCs

The sequence surrounding the mutations in the *DNMT3B* gene was determined for ICF iPSCs pR and pG to ensure the proper design of single guide RNAs (sgRNA) and single stranded oligodeoxynucleotides (ssODNs). sgRNAs were designed using the Optimized CRISPR Design online tool (http://crispr.mit.edu) and cloned into pSpCas9(BB)−2A-Puro (PX459) V2.0, a gift from Feng Zhang (Addgene plasmid #62988; http://www.addgene.org/62988/; RRID:Addgene_62988) (*Ran et al., 2013*). sgRNA were designed for the following sequences: pR: 5'-GGCTTTCCTGTGCACTACAC-3'; pG: 5'-CTGGAGGCTAATCCGCACCC-3'. The template ssODNs included: pR: 5'- ACGCTCCAG-GACCTTCCCAGCAGCTTCTGGCGGGCACCACGGCCCATGTTGGACACGTCTGTATAGTGCACAG-GAAAGCCAAAGATCCTG-3'; pG: 5'-CGCTCACTATGTCAGCGCCTGACACCACCCCAGTCCCCAC TGCCCAGCCACCTCTGATCTCGGGGGTCCGGATAGCCTCCAGGATTGGGG-3'. Details relevant to sgRNA and ssODN design are described in *Figure 1—figure supplement 1*. ssODNs were diluted in 10 mM Tris HCl, pH8.0 to a concentration of 100 µM.

The sgRNA/Cas9 plasmid and the ssODN were introduced into the iPSCs by electroporation. The day prior to electroporation, iPSCs ICF were fed with media containing 5 µM ROCK inhibitor (Y-27632, StemCell Technologies) and DR4, multidrug-resistant mouse embryonic fibroblasts (MEFs) (*Tucker et al., 1997*) were plated on a gelatinized 10 cm tissue culture dish. Immediately prior to electroporation, the cells were dissociated with Accutase (07920, StemCell Technologies) and 0.5−2

**Table 1.** Induced pluripotent stem cells (iPSCs) utilized in this study

| iPSC name | Additional information | Reference |
|---|---|---|
| pR | pR75 – original ICF1 iPSC | (*Sagie et al., 2014*) |
| cR7 | cR7 – CRISPR corrected pR clone | this manuscript |
| cR35 | cR35 – CRISPR corrected pR clone | this manuscript |
| pG | pG20 – original ICF1 iPSC | (*Sagie et al., 2014*) |
| cG13 | cG13 – CRISPR corrected pG clone | this manuscript |
| cG50 | cG50 – CRISPR corrected pG clone | this manuscript |
| WT1 | UN1-22 – derived from a healthy individual | (*Shinnawi et al., 2015*) |
| WT2 | FSE-5m – derived from a healthy individual | (*Yehezkel et al., 2011*) |

× 10^6 cells were washed with Opti-MEM (31985070, ThermoFisher SCIENTIFIC, MA, USA) and re-suspended in 100 µl of Opti-MEM containing 10 µg of the sgRNA/Cas9 plasmid and 1 µL of 100 µM ssODN, per 1 × 10^6 cells. Cells were electroporated using NEPA21 Electroporator (NEPAGENE, Chiba, Japan) using the following parameters: voltage - 125 mV, pulse length - 5 ms, interval length – 50 ms, number of pulses - 2, decay rate - 40%. Following electroporation, cells were plated on the previously prepared DR4 MEFs and fed with SR medium containing Knockout DMEM (10829018, ThermoFisher SCIENTIFIC) supplemented with 20% Knockout serum replacement (10828028, ThermoFisher SCIENTIFIC), 4 ng/ml recombinant Human FGF-basic (100-18B, PeproTech), 1% nonessential amino acids (11140035, ThermoFisher SCIENTIFIC), 0.1 mM β-mercaptoethanol (31350010, ThermoFisher SCIENTIFIC) and 1 mM glutamine (A2916801, ThermoFisher SCIENTIFIC), and supplemented with 5 µM of ROCK inhibitor and 5 µM L-755507 (73992, StemCell Technologies), which was proposed to improve the efficiency of HDR in human induced-pluripotent stem cells (*Yu et al., 2015*). Twenty-four hours following electroporation, selection was applied with 1 µg/ml Puromycin for 48 hr (together with ROCK inhibitor and L-755507).

Approximately fifty clones were screened for each patient iPSCs. HDR with the ssODNs incorporated a restriction site to facilitate the detection of positive clones by RFLP (*Figure 1—figure supplement 1*). Genomic DNA from individual clones was amplified (pR screening primers: pR-Forward 5'-ATGGCGAGGGCAGAAAGAGT-3', pR-Reverse- 5'-GCACCGTGTTAGGCTGCTCC-3'; pG screening primers: pG-Forward- 5'-GCCTGTCCACATGGAACC-3', pG-Reverse- 5'-CTAGAGCTCTCTTCCC TCC-3') and digested with the appropriate restriction enzyme (*AflIII* for pR – naturally occurring in WT DNA, and *AvaII* for pG) (*Figure 1—figure supplement 1*). During the screening for corrected pG clones, we ensured that DNA extracted from the selected clones was amenable to digestion following PCR by amplifying a genomic region located in intron 1 of *DNMT3B* which harbors an *AvaII* site (primers for the control region: Forward: 5'- CTGCTCCAATGCTGCCCC-3', Reverse 5'-GGAGGGCGGATTACATGAGG-3') and digesting with *AvaII*. Clones which presented with the appropriate digestion pattern were Sanger sequenced to validate the correction.

## Southern analyses of telomere length and of DNA methylation

Genomic DNA was extracted according to standard procedures. TRF analysis for telomere length determination was performed on *HinfI*-digested DNA as described in *Yehezkel et al. (2008)*. To determine subtelomeric methylation, a modified TRF analysis was performed by digesting DNA with each of the following isoschizomeric enzymes: *HpaII* (methylation sensitive enzyme) and *MspI* (methylation insensitive enzyme). Digested DNA was separated on a 0.7% agarose gel and transferred to a charged nylon transfer membrane (1226556, GVS North America, ME, USA) using a BIO-RAD vacuum blotter. Due to a gradient in the degree of separation along the gel width, a size marker was run at both sides of the gel. Membranes were hybridized to a C-rich telomeric probe and washed as described (*Yehezkel et al., 2008*). Methylation analyses of NBL-1, satellite 2 and p1A12 repeats were performed as previously described (*Ofir et al., 2002*; *Yehezkel et al., 2011*).

## Whole-Genome bisulfite sequencing (WGBS) analysis

BS-seq libraries were generated from 100 ng genomic DNA using the TruSeqDNA Methylation kit (15066014, Illumina, CA, USA) and sequenced to obtain 160 million paired-end reads (2 × 100 bp) at Genomix4life (Italy) using the Illumina HiSeq2500 platform. Read quality was analyzed using FastQC tool. Adaptors and low quality bases were trimmed out using Cutadapt (settings: -u 7, -U 7, -m 40, -q 30, –trim-n). Reads were then aligned to the customized reference human genome GRCh38 (Release 86) available on the Ensembl FTP, using the Bismark aligner (v0.14.5, settings: –score_min L,0,–0.6 -X 1000 -I 0). Methylation calls were performed using Bismark methylation extractor script. We used a R homemade script to calculate methylation percentages per each single CpG, by dividing the number of methylated Cs by the total coverage of that base. CpGs with a coverage of between 4x to 100x were retained for CpG methylation analysis.

Genomic coordinates of the subtelomeric regions, as well as those of HSATII, GSATII, NBL-1 and p1A12 repeats were obtained by aligning their consensus sequences to the hg38 genome using BLAT software (University of California, Santa Cruz [UCSC] Genome Browser, http://genome.ucsc. edu). High confidence subtelomeric sequences were obtained from *Stong et al. (2014)*. The HSATII consensus sequence was downloaded from the Repbase Update database (*Bao et al., 2015*), and

the GSATII coordinates were directly extracted from the file hg38 repeamasker_clean.txt.gz available as annotation of the RepEnrich tool (*Criscione et al., 2014*). The NBL-1 and p1A12 repeat sequences (*Brock et al., 1999*; *Thoraval et al., 1996*) were downloaded from NCBI GenBank (U53226 and AF157964, respectively). Coordinates of the TERRA promoter repeats 29-, 37- and 61 bps were obtained by aligning the sequences reported in *Supplementary file 2* (based on *Brown et al., 1990*) to the subtelomeric sequences using the MAFFT v7 available in Benchling (Molecular Biology software, 2018). To graphically represent the position of the TERRA repeats in the subtelomere maps (reported in *Figure 3A*, *Figure 3—figure supplement 1* and *Figure 7—figure supplement 1*), the counts of each repeat within a bin were calculated by determining the bp length within the bin that overlapped with the repeat consensus. A repeat was counted within a bin only when at least 25% of the repeat consensus length overlapped the bin.

CpG counts were analyzed using cpgDensityCalc function, available in Repitools R package. Heatmaps and hierarchical clustering were produced using heatmap2 function in R 3.03 included in the gplots package. For intersection analysis and management of genomic coordinates Bedtools v2.17.0 (*Quinlan and Hall, 2010*) were used.

## Methylation analysis by targeted bisulfite sequencing

0.5–1 µg of genomic DNA was bisulfite-converted with the Methylamp DNA Modification kit (EPIGENTEK, NY, USA). After the conversion, DNA was amplified using FastStart Taq DNA Polymerase, dNTPack (04738314001, MERCK, Darmstadt, Germany) with the following program: 95°C for 5 min; 4 cycles of 95°C for 1 min, 53°C for 3 min, 72°C for 3 min; 2 cycles of 95°C for 30 s, 55°C for 45 s, 72°C for 45 s; 40 cycles of 95°C for 30 s, 72°C for 1.5 min; 72°C for 10 min. PCR products were purified by QIAquick PCR purification kit (28104, QIAGEN, MD, USA). Further amplicon processing, sequencing, and analysis were performed as described (*Sagie et al., 2017b*). Primers for amplification of bisulfide converted DNA are described in *Supplementary file 1*. In the case in which a primer set amplified several subtelomeres, we used sequence differences between the amplified subtelomeres to identify the origin of the PCR product (*Supplementary file 1*, footnote).

## RNA isolation and quantitative RT-PCR (RT-qPCR)

RNA was extracted using the RNeasy Mini Kit (74104, QIAGEN), followed by purification with TURBO DNA-*free* TM Kit (AM2238, ThermoFisher SCIENTIFIC). 1 µg of RNA was reverse transcribed at 55°C using SuperScript III reverse transcriptase (18080044, ThermoFisher SCIENTIFIC), with a TERRA-specific primer composed from five telomere-hexameric repeats $(CCCTAA)_5$, and a β-actin-specific primer (5'-AGTCCGCCTAGAAGCATTTG-3') as described (*Sagie et al., 2017b*). Expression levels of subtelomeric-specific TERRA were determined by RT-qPCR using specific subtelomere primers (described in *Supplementary file 3*) normalizing to the β-actin gene by the delta delta Ct method. RT-qPCR was done with Fast SYBR Green Master Mix (4385612, ThermoFisher SCIENTIFIC), on an Applied Biosystems StepOnePlus Real-Time PCR system (AB-4385612, ThermoFisher SCIENTIFIC).

## Western analysis of DNMT3B

Cells were washed three times with phosphate buffered solution (PBS) and lysed with lysis buffer (100 mM Tris-HCl (pH8), 140 mM NaCl, 0,2% SDS, 1% NP40, 20 mM EDTA, containing protease inhibitor cocktail tablets (11697498001, MERCK). Protein lysates were quantified by Bradford protein assay (Bio-Rad). 30 µg of total protein lysates were separated by SDS polyacrylamide gel electrophoresis (SDS-PAGE) and transferred to PVDF membranes (Millipore), blocked with 2% BSA in TBS buffer with 0.1% Tween20. DNMT3B was detected with anti-DNMT3B diluted 1:3000 (C15410218, Diagenode). For protein loading normalization, membranes were also reacted with an anti-actin antibody diluted 1:5000 (A2066, Sigma). Protein levels were calculated by using Typhoon Scan and ImageQuant 5.2 software.

## Derivation of Fibroblast-Like cells (FLs) from iPSCs

IPSCs were induced to differentiate through generation of embryoid bodies (EBs), as described previously (*Yehezkel et al., 2011*). Following 10 days in suspension, EBs were seeded on gelatin-coated 6-well culture plates, as described (*Yehezkel et al., 2011*). Differentiating cultures were grown for at

least two additional weeks. FLs were isolated based on their morphology, as described (*Yehezkel et al., 2011*).

## Ectopic expression of human TERT in FL cells

pR C7 FLs were serially infected 2–4 times with the lentiviral vector pLOX-TERT-iresTK (a gift from Didier Trono, Addgene plasmid # 12245; http://www.addgene.org/12245/; RRID:Addgene_12245) (*Salmon et al., 2000*). Following infection, the FLs were grown as described above. PDs were recorded at each passage.

## Chromatin immunoprecipitation (ChIP)

$25 \times 10^6$ cells were treated with 1% formaldehyde for 10 min at room temperature. To arrest the crosslinking, glycine was added to 0.125 M. Fixed cells were pelleted by centrifugation at 780 g for 5 min at 4°C. The cells were rinsed with ice cold phosphate buffered solution (PBS) and pelleted by centrifugation at 780 g for 5 min at 4°C. Cells were then resuspended in 6 ml of lysis buffer (100 mM PIPES pH 8.0, 85 mM KCl, 0.5% NP40 and protease inhibitor cocktail (11873580001, MERCK)), centrifuged at 780 g for 5 min at 4°C and washed in cold PBS by additional centrifugation under the same conditions. Cells were then resuspended in 1 ml of nuclei lysis buffer (50 mM Tris-HCl pH 8.0, 10 mM EDTA, 0.8% sodium dodecyl sulfate (SDS) and protease inhibitor cocktail) and sonicated (using the Bioruptor UCD-200, Diagenode, NJ, USA) under conditions that gave a range of DNA fragments between 200–1000 bp. Following sonication, the lysed chromatin was centrifuged at 9300 g to eliminate debris, and the supernatant containing the soluble chromatin was separated to a new tube. ChIP for H3K4me3 and H3K9me3 was performed by incubating 80 µg of chromatin diluted in 1 ml of ChIP dilution buffer (1% Triton X-100, 0.5 mM EGTA, 10 mM Tris-HCl, pH 8.1, 140 mM NaCl and protease inhibitor cocktail) with 3.5 µg of each of the antibodies (H3K4me3: ab8580, Abcam, Cambridge, UK, H3K9me3: ab8898,Abcam). ChIP for H3K36me3 was carried out with 5 µg of antibody (ab9050, Abcam). Immunoprecipitated complexes were recovered with protein A/G PLUS-Agarose (sc-2003, Santa Cruz, CA, USA), washed with high salt buffer (PBS, 1% NP-40, 0.5% Sodium deoxycholate and 0.1% SDS) and wash buffer (100 mM, 500 mM, 1% NP-40 and 1% Sodium deoxycholate), reverse crosslinked by resuspending the beads in 200 µl of 1% SDS, 0.1 M Sodium bicarbonate and 200 mM NaCl at 67°C for two hours. Following centrifugation, the supernatant was incubated with proteinase K mix (final concentrations: 0.2 mg/ml proteinase K, (03115828001, MERCK), 200 mM Tris HCl, pH 6.8 and 100 mM EDTA, pH 8.0) at 67°C for a minimum of 3 hr and purified by QIAquick PCR purification kit (28106, QIAGEN). To determine enrichment of DNMT3B binding, chromatin from iPSCs was immunoprecipitated with 5 µg of anti-DNMT3B (Diagenode Cat # C15410218) using Magna ChIP (17–610, MERCK). qPCR was performed on the immunoprecipitated samples using primers described in *Supplementary file 3*. Results were expressed as site occupancy (normalization was done relative to an amplified negative region in the same sample) to exclude technical variations between samples. At first, a normalization to the input amount was calculated and then, the obtained percentage of input was used to calculate the enrichment over the percentage of input of a negative control region (Hoxa7 TSS, GAPDH promoter, or Myoglobin exon 2; *Supplementary file 3*).

## Treatment with OICR-9429

OICR-9429 (SML1209, MERCK) was dissolved in DMSO at a stock concentration of 1 mM. For drug treatment, 1 mM OICR-9429 or DMSO were dissolved in pre-warmed media at a 1:1000 ratio to achieve a final concentration of 1 µM of OICR-9429 and 0.1% DMSO. iPSCs were treated for 72 hr, replacing the medium every 24 hr. Following the treatment, iPSCs were harvested and a H3K4me3 ChIP assay was performed, as described above. DNA methylation levels in the treated and control samples were evaluated by targeted bisulfite DNA sequencing of subtelomeres, with at least 750 reads analyzed for each CpG.

## SA-β-Gal assay

FLs were grown to senescence and were subjected to SA-β-Gal staining at pH 6, as described in *Dimri et al. (1995)* and *Shlush et al. (2011)* or with the Senescence beta-Galactosidase Staining Kit

(9860S, Cell Signaling TECHNOLOGY). Images of stained cells were captured using a Nikon ECLIPSE Ti inverted microscope.

## Statistical analysis

All statistical analyses were performed using Python 2.7 and the Python libraries 'numpy', 'scipy', 'statsmodels', 'pandas', 'jupyter' and 'notebook'. 'Matplotlib' and 'seaborn' were used for graphics. Comparison of WGBS methylation data between samples was done by calculating the effect size of sample pairs using Cohen's *d* (*Sawilowsky, 2009*). For comparison of more than two samples in the targeted bisulfite experiments, one-way ANOVA test was carried out to test variance of the mean methylation levels between samples. Tukey's HSD post hoc test was done to determine which sample pairs show a significant difference in level of methylation. For TERRA and ChIP analyses, bars and error bars represent means and ± standard error of means (SEM), respectively, based on at least three independent experiments. Statistical analyses were done using the non-parametric Mann-Whitney U-test. For the ChIP experiments following OICR treatment, statistical analyses were performed using a two-tailed Student's t-test. Statistical analysis for targeted bisulfite experiments following OICR treatment was done using the non-parametric Mann-Whitney U-test on the methylation delta value of each CpG.

## Data availability statement

All sequencing data generated as part of this work are publicly available through NCBI Gene Expression Omnibus (GEO). Whole genome bisulfite sequencing dataset and targeted bisulfite sequencing datasets are under accession numbers GSE137183 and GSE138265, respectively. The scripts used for WGBS and PGM data analyses have been deposited at: https://github.com/Shitou19/methylation_analysis (*Toubiana, 2019*; copy archived at https://github.com/elifesciences-publications/methylation_analysis).

## Acknowledgements

We are grateful to Marc Lalande and Christopher Stoddard for their advice and help in the genome editing procedures. We thank Claudia Angelini for her help with bioinformatics and statistic tools, and Noam Hershtig for his help with statistical analyses. Many thanks to Daniel Kornitzer and Aya Tzur-Gilat for comments on the manuscript. Shir Toubiana is grateful to The Edmond de Rothschild Foundation (IL) for funding her PhD scholarship.

## Additional information

### Funding

| Funder | Grant reference number | Author |
|---|---|---|
| Israel Science Foundation | 1362/17 | Sara Selig |
| Telethon | GGP15209 | Maria R Matarazzo |
| PON/MISE | 2014-2020 FESR F/050011/01-02/X32 | Maria R Matarazzo |
| MIUR/CNR | Epigenomics Flagship Project (EPIGEN) | Maria R Matarazzo |

The funders had no role in study design, data collection and interpretation, or the decision to submit the work for publication.

### Author contributions

Shir Toubiana, Data curation, Software, Formal analysis, Validation, Investigation, Visualization, Methodology, Writing—original draft, Writing—review and editing; Miriam Gagliardi, Data curation, Software, Formal analysis, Validation, Investigation, Visualization, Methodology, Writing—review and editing; Mariarosaria Papa, Formal analysis, Investigation; Roberta Manco, Formal analysis, Validation, Investigation, Visualization; Maty Tzukerman, Investigation, Writing—review and editing; Maria

R Matarazzo, Sara Selig, Conceptualization, Supervision, Funding acquisition, Validation, Writing—original draft, Project administration, Writing—review and editing

**Author ORCIDs**
Shir Toubiana (iD) https://orcid.org/0000-0003-3001-5281
Miriam Gagliardi (iD) https://orcid.org/0000-0003-0156-3705
Mariarosaria Papa (iD) http://orcid.org/0000-0002-2467-4187
Roberta Manco (iD) https://orcid.org/0000-0003-2888-3442
Maty Tzukerman (iD) https://orcid.org/0000-0002-9080-2511
Maria R Matarazzo (iD) https://orcid.org/0000-0002-8192-4322
Sara Selig (iD) https://orcid.org/0000-0001-5759-9948

**Decision letter and Author response**
Decision letter https://doi.org/10.7554/eLife.47859.sa1
Author response https://doi.org/10.7554/eLife.47859.sa2

## Additional files

**Supplementary files**
• Supplementary file 1. Primers for bisulfite analysis by Ion torrent PGM.
• Supplementary file 2. TERRA promoter consensus repeats utilized for alignments with WGBS data.
• Supplementary file 3. Primers for RT-qPCR and ChIP analyses.
• Transparent reporting form

**Data availability**
Sequencing data have been deposited in GEO under accession codes GSE137183 and GSE138265. The scripts used for WGBS and PGM data analyses have been deposited at: https://github.com/Shitou19/methylation_analysis (copy archived at https://github.com/elifesciences-publications/methylation_analysis).

The following datasets were generated:

| Author(s) | Year | Dataset title | Dataset URL | Database and Identifier |
|---|---|---|---|---|
| Toubiana S, Gagliardi M, Papa M, Tzukerman M, Matarazzo MR, Selig S | 2019 | Persistent epigenetic memory impedes rescue of the telomeric phenotype in human ICF iPSCs following DNMT3B correction | https://www.ncbi.nlm.nih.gov/geo/query/acc.cgi?acc=GSE137183 | NCBI Gene expression Omnibus, GSE137183 |
| Toubiana S, Gagliardi M, Papa M, Tzukerman M, Matarazzo MR, Selig S | 2019 | Persistent epigenetic memory impedes rescue of the telomeric phenotype in human ICF iPSCs following DNMT3B correction | https://www.ncbi.nlm.nih.gov/geo/query/acc.cgi?acc=GSE138265 | NCBI Gene expression Omnibus, GSE138265 |

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
