## [Decision Letter]

**Acceptance summary:**

The manuscript by Selig and colleagues addresses the important question of whether epigenetic defects caused by mutations in epigenetic regulators can be restored after gene correction. The authors use patient specific iPSCs to show that repair of the DNMT3B mutation that causes human ICF1 syndrome is unable fully restore the ICF1-related epigenetic abnormalities those cells. These findings are important for anyone interested in developing translational approaches in disorders such as ICF1. When using patient-specific cells for either iPSC-based disease modeling or for regenerative approaches, gene-corrected cells will likely require additional interventions to restore the persistent epigenetic defects. The work is also of intriguing for the community interested in basic aspects of DNA methylation, as it demonstrates a remarkable specificity of the methylation defects restricted to repeats at subtelomeric but not pericentromeric regions. Furthermore, high expression of the long non-coding RNA TERRA and the functional link of H3K4 methylation to the persistent methylation defects point to a mechanistic hierarchy and define key restriction factors for establishing proper methylation levels. Finally, the study raises the question of whether this finding will apply to other disease models affecting epigenetic regulators. It will also be intriguing to test whether specific pluripotent states such as the use of naïve rather than primed state human iPSCs may overcome the observed methylation defects in combination or instead of strategies that directly inhibit H3K4 methylation.

**Decision letter after peer review:**

Thank you for submitting your article "Persistent epigenetic memory impedes rescue of the telomeric phenotype in human ICF iPSCs following DNMT3B correction" for consideration by *eLife*. Your article has been reviewed by two peer reviewers, and the evaluation has been overseen by a Reviewing Editor and Huda Zoghbi as the Senior Editor. The following individuals involved in review of your submission have agreed to reveal their identity: Dirk Hockemeyer (Reviewer #1), Lea Harrington (Reviewer #2).

The reviewers have discussed the reviews with one another and the Reviewing Editor has drafted this decision to help you prepare a revised submission.

This manuscript addresses the dynamics of DNA methylation by DNMT3B during early development using a human ICF syndrome iPSC line and its gene corrected, isogenic counterpart. The work demonstrates that in the gene-corrected iPSC line pericentromeric but not subtelomeric regions restore normal methylation patterns and that the persistent defects in subtelomeric methylation are associated with persisting disease phenotypes in iPSC-derived fibroblasts. Interestingly, subtelomeric regions resistant to de novo methylation are characterized by high H3K4me3 levels and expression of the telomeric repeat-containing long non-coding RNA TERRA. Finally, the authors demonstrate that more complete subtelomeric methylation cells can be achieved following pharmacological inhibition of H3K4me3 in gene-corrected iPSCs.

Overall, all the reviewers found the work of considerable interest and the findings intriguing. There were however, several points that should be addressed in a revised manuscript as detailed below.

Essential revision:

1) Confirming proper expression levels for DNMT3B in the gene corrected clones: it would be important to demonstrate that the levels of DNMT3B were back to physiological levels after gene correction and thereby to rule out partial rescue of levels.

2) Telomere length-dependent senescence of DNMT3B-restored fibroblasts: We encourage the authors to perform TERT overexpression experiments (as in their past studies) to address whether TERT expression can rescue senescence phenotype in the differentiated fibroblasts. The reviewers think that show experiments can be completed within the 2 months period, and that either result would be a valuable addition to further strengthen the manuscript.

3) Data consistency about telomere length: The authors should discuss / explain the apparent discrepancy in the telomere length data in Figure 5D vs Figure 1D (for example in Figure 1D, the same corrected clones have shorter telomeres than the parental iPSCs).

4) Discussing limitations of the model system: The hiPSC cultures used in these studies represent a relatively late (epiblast) state of development that may be too late for achieving full remethylation. There is currently no ideal system; the use of a "naïve" hiPSC culture system might be of uncertain value given the controversies about the proper conditions for naive pluripotency in the human system. Furthermore, such studies would be too extensive as a revision request. Nevertheless, we think it is important for the authors to discuss the limitations of the current system used, as it could impact the conclusions about translation – e.g. the possibility of achieving more complete remethylation when using a more naïve pluripotent culture condition and derivatives from those more naïve cells in the future.

---

## [Author Response]

Essential revision:1) Confirming proper expression levels for DNMT3B in the gene corrected clones: it would be important to demonstrate that the levels of DNMT3B were back to physiological levels after gene correction and thereby to rule out partial rescue of levels.

We previously reported DNMT3B protein levels in ICF iPS pG prior to CRISPR correction. In this iPS we expected a priori a reduction in protein levels due to the premature stop codon in one of the DNMT3B alleles which could lead to nonsense mediated decay. However, despite this mutation, protein levels seemed normal. We published this in Sagie et al., 2014 in Supplemental Figure S2A. In ICF iPS pR which are homozygous to a missense mutation, we did not necessarily expect significantly lower protein levels, although missense mutations could theoretically destabilize the protein.

To ensure that indeed the protein levels in all of the corrected clones are within a normal range, we performed western analysis with an anti-DNMT3B antibody. We focused our analysis on the 3B1 and 3B2 isoforms which are the isoforms that contain the catalytic domain of DNMT3B. For normalization of the loaded protein levels we used an anti-actin antibody. The results of this analysis indicate that DNMT3B1+2 protein levels in the isogenic ICF iPSCs are equal or higher than those of WT-iPSCs. We are therefore confident that the partial rescue of methylation at subtelomeres is not a consequence of low levels of corrected DNMT3B.

We added a sentence in the manuscript describing these findings, and a figure displaying the western analysis (Figure 1—figure supplement 2).

2) Telomere length-dependent senescence of DNMT3B-restored fibroblasts: We encourage the authors to perform TERT overexpression experiments (as in their past studies) to address whether TERT expression can rescue senescence phenotype in the differentiated fibroblasts. The reviewers think that show experiments can be completed within the 2 months period, and that either result would be a valuable addition to further strengthen the manuscript.

We thank the reviewers for suggesting this experiment. Its results indeed strengthen the conclusions that premature senescence in the corrected FLs is due to accelerated telomere shortening.

To note, the frozen FLs did not thaw well because they were already close to senescence when frozen. However, cR7 FLs, derived from the corrected pR7 iPSCs did partially survive the thawing and we succeeded in immortalizing them with a lentiviral plasmid containing hTERT. We determined the population doubling of the infected cells during continuous culturing for approximately 40 days, and compared the growth curve with that of non-infected cR7 FLs. While the non-infected cells entered senescence after approximately a month, the hTERT-expressing cR7 cells were rescued from premature senescence and continued to proliferate for over 40 days. SA-β-Gal staining confirmed that the non-infected culture entered senescence, but not the hTERT-expressing culture.

The results of this experiment are described in the manuscript in the paragraph starting with “Fibroblast-like cells derived from the corrected ICF iPSCs enter premature senescence”, and are shown in Figure 5—figure supplement 2.

3) Data consistency about telomere length: The authors should discuss / explain the apparent discrepancy in the telomere length data in Figure 5D vs Figure 1D (for example in Figure 1D, the same corrected clones have shorter telomeres than the parental iPSCs).

We would like to respond to two issues:

a) The finding that corrected clones have shorter telomeres than the parental iPSCs reflects a phenomenon described previously by several groups, including ours. iPSC clones may show different telomere length dynamics during prolonged growth in culture, even if they have a common parental origin. As far as we are aware, the mechanism behind this phenomenon is not understood. We now refer to this point in the Results section when describing the TRF analysis in Figure 1D.

b) Apparent discrepancy in telomere length data in Figure 5D vs Figure 1D: we carefully checked the two blots and can explain the source of this apparent discrepancy. We first want to point out that in Figure 1, the samples are digested with *MspI*, while in Figure 5 they are digested with *HinfI*. We performed a Southern analysis that confirmed that the TRF sizes of several DNA samples are very similar when digested with each of these two enzymes.

One factor that can partially explain the apparent inconsistency in telomere lengths of samples from equal or very similar passages lies in the different extent of DNA separation of both gels. The DNA in the gel that produced Figure 1D was separated over a longer distance than that of 5D and therefore the molecular weights between 4-10kb have much better separation, emphasizing the lower molecular weight smear. Another factor we recently discovered that influences the molecular weights of the TRF hybridization smear is related to a technical issue that arises during long-range electrophoreses of DNA: when separating the DNA on a very wide and large apparatus, a slight defect in the uniformity of the electric field causes a gradient to form in the extent of separation across the width of the gel. The DNA in the lanes positioned nearer the side of the apparatus that contains the contact point of the cathode wire migrates slightly faster. In the original figures, we marked the Southerns according to the size markers in the lane furthest away from the cathode wire. However these marker sizes do not represent accurately the sizes at the other end of the gel. To ensure that this indeed explains the alleged discrepancy, we quantitated telomere lengths using “MATELO” software (Yehezkel et al., 2011), this time using the size markers closest to the analyzed lanes. Author response table 1 demonstrates that when quantified in this manner, the mean telomere lengths of each of the samples are very similar between both gels. In the table, we designate the marker to which each sample was aligned to.

**Author response table 1. resptable1:** 

	Mean Telomere Length
MspI-*digested (Figure 1D)*	HinfI-*digested (Figure 5D)*
pR	Left marker – 7.2 Kb	Left marker – 7.00 Kb
cR7	Left marker – 6.4 Kb	Left marker – 6.9 Kb
cR35	Right marker – 6.6 Kb	Left marker – 6.7 Kb
WT1	Right marker – 10.0 Kb	Right marker – 10.4 Kb

To provide a more accurate estimation of molecular sizes, we now include in all the TRF blots a size marker at each side of the gel. We also explain this point briefly in the legends of the relevant figures.

4) Discussing limitations of the model system: The hiPSC cultures used in these studies represent a relatively late (epiblast) state of development that may be too late for achieving full remethylation. There is currently no ideal system; the use of a "naïve" hiPSC culture system might be of uncertain value given the controversies about the proper conditions for naive pluripotency in the human system. Furthermore, such studies would be too extensive as a revision request. Nevertheless, we think it is important for the authors to discuss the limitations of the current system used, as it could impact the conclusions about translation – e.g. the possibility of achieving more complete remethylation when using a more naïve pluripotent culture condition and derivatives from those more naïve cells in the future.

We now discuss this point in more details in the Discussion. We quote studies that show that in naïve stem cells the chromatin acquires a more open conformation in comparison to primed stem cells, including lower DNA methylation levels, lower H3K9me3 and higher H3K4me3 at CpG-rich promoters. We therefore do not anticipate that reverting the cells to a naïve state will necessarily rescue the methylation defects at subtelomeres. However, we state in the discussion that: “However, we cannot exclude that H3K4me3 levels may show a distinct profile at TERRA promoters in the naïve state, and in addition, that a varying repertoire of transcription factors could alleviate the permissive epigenetic landscape, allowing the binding of DNMT3B”.